# StyleDrop: Text-to-Image Generation in Any Style

**Kihyuk Sohn    Nataniel Ruiz    Kimin Lee**\*    **Daniel Castro Chin    Irina Blok**

**Huiwen Chang**[†]    **Jarred Barber    Lu Jiang    Glenn Entis    Yuanzhen Li**

**Yuan Hao    Irfan Essa    Michael Rubinstein    Dilip Krishnan**

Google Research

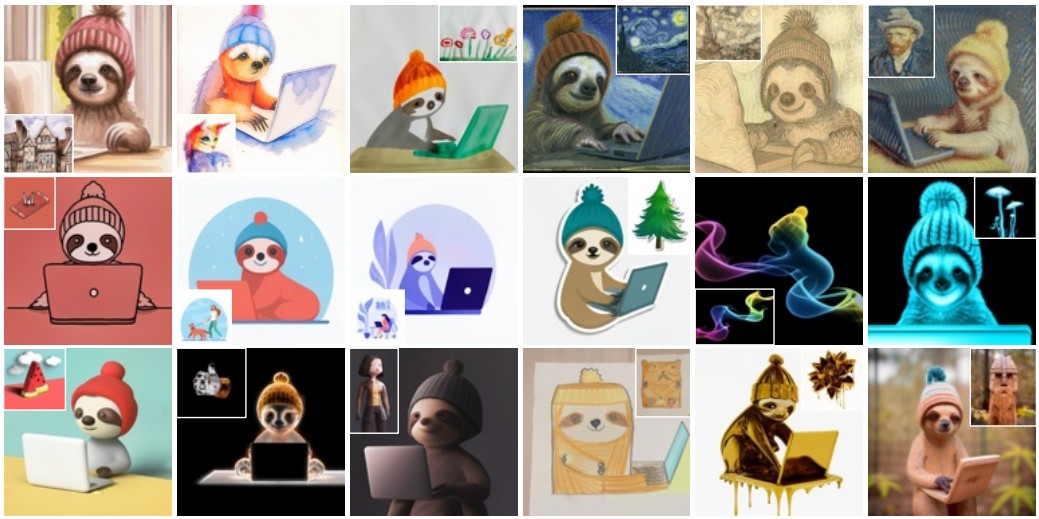

Figure 1: **Visualization of StyleDrop** outputs for 18 different styles. Each model is tuned on a *single* style reference image, which is shown in the white insert box of each image. The per-style text descriptor is appended to the content text prompt: "*A fluffy baby sloth with a knitted hat trying to figure out a laptop, close up*". Generated images capture many nuances such as colors, shading, textures and 3D appearance.

## Abstract

Pre-trained large text-to-image models synthesize impressive images with an appropriate use of text prompts. However, ambiguities inherent in natural language and out-of-distribution effects make it hard to synthesize image styles, that leverage a specific design pattern, texture or material. In this paper, we introduce *StyleDrop*, a method that enables the synthesis of images that faithfully follow a specific style using a text-to-image model. The proposed method is extremely versatile and captures nuances and details of a user-provided style, such as color schemes, shading, design patterns, and local and global effects. It efficiently learns a new style by fine-tuning very few trainable parameters (less than $1\%$ of total model parameters) and improving the quality via iterative training with either human or automated feedback. Better yet, StyleDrop is able to deliver impressive results even when the user supplies only a *single* image that specifies the desired style. An extensive study shows that, for the task of style tuning text-to-image models, StyleDrop implemented on Muse [5] convincingly outperforms other methods, including DreamBooth [34] and textual inversion [11] on Imagen [35] or Stable Diffusion [33]. More results are available at our project website: https://styledrop.github.io.

---
\*Now at Korea Advanced Institute of Science and Technology (KAIST).
[†]Now at OpenAI.

37th Conference on Neural Information Processing Systems (NeurIPS 2023).

# 1 Introduction

Text-to-image models trained on large image and text pairs have enabled the creation of rich and diverse images encompassing many genres and themes [2, 5, 33, 35, 43]. The resulting creations have become a sensation, with Midjourney [2] reportedly being the largest Discord server in the world [1]. The styles of famous artists, such as Vincent Van Gogh, might be captured due to the presence of their work in the training data. Moreover, popular styles such as "anime" or "steampunk", when added to the input text prompt, may translate to specific visual outputs based on the training data. While many efforts have been put into "prompt engineering", a wide range of styles are simply hard to describe in text form, due to the nuances of color schemes, illumination and other characteristics. As an example, Van Gogh has paintings in different styles (*e.g.*, Fig. 1, top row, rightmost three columns). Thus, a text prompt that simply says "Van Gogh" may either result in one specific style (selected at random), or in an unpredictable mix of several styles. Neither of these is a desirable outcome.

In this paper, we introduce StyleDrop[3] which allows significantly higher level of stylized text-to-image synthesis, using as few as *one* image as an example of a given style. Our experiments (Fig. 1) show that StyleDrop achieves unprecedented accuracy and fidelity in stylized image synthesis. StyleDrop is built on a few crucial components: (1) a transformer-based text-to-image generation model [5]; (2) adapter tuning [15]; and (3) iterative training with feedback. For the first component, we find that Muse [5], a transformer modeling a discrete visual token sequence, shows an advantage over diffusion models such as Imagen [35] and Stable Diffusion [33] for learning fine-grained styles from single images. For the second component, we employ adapter tuning [15] to style-tune a large text-to-image transformer efficiently. Specifically, we construct a text input of a style reference image by composing content and style text descriptors to promote content-style disentanglement, which is crucial for compositional image synthesis [37, 32, 41]. Finally, for the third component, we propose an iterative training framework, which trains a new adapter on images sampled from a previously trained adapter. We find that, when trained on a small set of high-quality synthesized images, iterative training effectively alleviates overfitting, a prevalent issue for fine-tuning a text-to-image model on a very few (*e.g.*, one) images. We study high-quality sample selection methods using CLIP score (*e.g.*, image-text alignment) and human feedback in Sec. 4.4.3, verifying the complementary benefit.

In addition to handling various styles, we extend our approach to customize not only style but also content (*e.g.*, the identifying/distinctive features of a given object or subject), leveraging DreamBooth [34]. We propose a novel approach that samples an image of *my content in my style* from two adapters trained for content and style independently. This compositional approach voids the need to jointly optimize on both content and style images [20, 13] and is therefore very flexible. We show in Fig. 5 that this approach produces compelling results that combines personalized generation respecting both object identity and object style.

We test StyleDrop on Muse on a diverse set of style reference images, as shown in Fig. 1. We compare with other recent methods including DreamBooth [34] and Textual Inversion [11], using Imagen [35] and Stable Diffusion [33] as pre-trained text-to-image backbones. An extensive evaluation based on prompt and style fidelity metrics using CLIP [29] and a user study shows the superiority of StyleDrop to other methods. Please visit our website and Appendix for more results.

# 2 Related Work

**Personalized Text-to-Image Synthesis** has been studied to edit images of personal assets by leveraging the power of pre-trained text-to-image models. Textual inversion [11] and Hard prompt made easy (PEZ) [39] find text representations (*e.g.*, embedding, token) corresponding to a set of images of an object without changing parameters of the text-to-image model.

DreamBooth [34] fine-tunes an entire text-to-image model on a few images describing the subject of interest. As such, it is more expressive and captures the subject with greater details. Parameter-efficient fine-tuning (PEFT) methods, such as LoRA [16] or adapter tuning [15], are adopted to improve its efficiency [3, 27]. Custom diffusion [20] and SVDiff [13] have extended DreamBooth to synthesize multiple subjects simultaneously. Inversion-based Style Transfer [44] presents a one-shot style tuning of text-to-image diffusion models. Unlike these methods built on text-to-image diffusion

---

[3]"StyleDrop" is inspired by eyedropper (*a.k.a* color picker), which allows users to quickly pick colors from various sources. Likewise, StyleDrop lets users quickly and painlessly 'pick' styles from a single (or very few) reference image(s), building a text-to-image model for generating images in that style.

models, we build StyleDrop on Muse [5], a generative vision transformer. [11, 39, 20] have shown learning styles with text-to-image diffusion models, but from a handful or a dozen of style reference images, and are limited to painting styles. We demonstrate on a wide variety of visual styles, including 3d rendering, design illustration, and sculpture, using a single style reference image.

**Neural Style Transfer (NST).** A large body of work [12, 18, 24, 7] has investigated style transfer using deep networks by solving a composite objective of style and content consistency [12]. Recently, [17] has shown that quantizing the latent space leads to improved visual and style fidelity of NST compared to continuous latent spaces. MaskSketch [4] converts sketch images into natural images via structure-guided parallel decoding of a masked image generation model. While both output stylized images, StyleDrop is different from NST in many ways; ours is based on text-to-image models to generate content, whereas NST uses an image to guide content (*e.g.*, spatial structure) for synthesis; we use adapters to capture fine-grained visual style properties; we incorporate feedback signals to refine the style from a single input image.

**Parameter Efficient Fine Tuning (PEFT)** is a new paradigm for fine-tuning of deep learning models by only tuning a much smaller number of parameters, instead of the entire model. These parameters are either subsets of the original trained model, or small number of parameters that are added for the fine-tuning stage. PEFT has been introduced in the context of large language models [15, 23, 16], and then applied to text-to-image diffusion models [35, 33] with LoRA [3] or adapter tuning [27]. Fine-tuning of autoregressive (AR) [10, 43, 21] and non-autoregressive (NAR) [6, 5, 38] generative vision transformers has been studied recently [36], but without the text modality.

## 3 StyleDrop: Style Tuning for Text-to-Image Synthesis

StyleDrop is built on Muse [5], reviewed in Sec. 3.1. There are two key parts. The parameter-efficient fine-tuning of a generative vision transformer (Sec. 3.2) and an iterative training with feedback (Sec. 3.3). Finally, we discuss how to synthesize images from two fine-tuned models in Sec. 3.4.

### 3.1 Preliminary: Muse [5], a masked Transformer for Text-to-Image Synthesis

Muse [5] is a state-of-the-art text-to-image synthesis model based on the masked generative image transformer, or MaskGIT [6]. It contains two synthesis modules for base image generation ($256 \times 256$) and super-resolution ($512 \times 512$ or $1024 \times 1024$). Each module is composed of a text encoder T, a transformer G, a sampler S, an image encoder E, and decoder D. T maps a text prompt $t \in \mathcal{T}$ to a continuous embedding space $\mathcal{E}$. G processes a text embedding $e \in \mathcal{E}$ to generate logits $l \in \mathcal{L}$ for the visual token sequence. S draws a sequence of visual tokens $v \in \mathcal{V}$ from logits via iterative decoding [6, 5], which runs a few steps of transformer inference conditioned on the text embeddings $e$ and visual tokens decoded from previous steps. Finally, D maps the sequence of discrete tokens to pixel space $\mathcal{I}$.[4] To summarize, given a text prompt $t$, an image $I$ is synthesized as follows:

$$I = \mathrm{D}\big(\mathrm{S}\left(\mathrm{G}, \mathrm{T}(t)\right)\big) \ , \ l_k = \mathrm{G}\left(v_k, \mathrm{T}(t)\right) + \lambda\big(\mathrm{G}\left(v_k, \mathrm{T}(t)\right) - \mathrm{G}\left(v_k, \mathrm{T}(n)\right)\big), \tag{1}$$

where $n \in \mathcal{T}$ is a negative prompt, $\lambda$ is a guidance scale, $k$ is the synthesis step, and $l_k$'s are logits, from which the next set of visual tokens $v_{k+1}$'s are sampled. We refer to [6, 5] for details on the iterative decoding process. The T5-XXL [30] encoder for T and VQGAN [10, 42] for E and D are used. G is trained on a large (image, text) pairs $\mathcal{D}$ using masked visual token modeling loss [6]:

$$L = \mathbb{E}_{(x,t)\sim\mathcal{D}, \mathbf{m}\sim\mathcal{M}}\Big[\mathrm{CE}_{\mathbf{m}}\big(\mathrm{G}\big(\mathrm{M}\left(\mathrm{E}(x), \mathbf{m}\right), \mathrm{T}(t)\big), \mathrm{E}(x)\big)\Big], \tag{2}$$

where M is a masking operator that applies masks to the tokens in $v_i$. $\mathrm{CE}_{\mathbf{m}}$ is a weighted cross-entropy calculated by summing only over the unmasked tokens.

### 3.2 Parameter-Efficient Fine-Tuning of Text-to-Image Generative Vision Transformers

Now we present a unified framework for parameter-efficient fine-tuning of generative vision transformers. The proposed framework is not limited to a specific model and application, and is easily applied to the fine-tuning of text-to-image (*e.g.*, Muse [5], Paella [31], Parti [43], RQ-Transformer [21]) and text-to-video (*e.g.*, Phenaki [38], CogVideo [14]) transformers, with a variety of PEFT methods, such as prompt tuning [23], LoRA [16], or adapter tuning [15], as in [36]. Nonetheless, we focus on Muse [5], an NAR text-to-image transformer, using adapter tuning [15].

---

[4]We omit the description of the super-resolution module for concise presentation, and point readers to [5] for a full description of the Muse model.

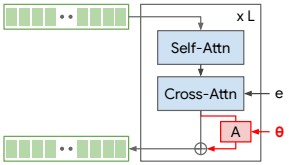

Figure 2: A simplified architecture of transformer layers of Muse [5] with modification to support parameter-efficient fine-tuning (PEFT) with adapter [15, 36]. $L$ layers of transformers are used to process a sequence of visual tokens in green conditioned on the text embedding $e$. Learnable parameters $\boldsymbol{\theta}$ are used to construct weights for adapter tuning. See Appendix B.1.1 for details on adapter architecture.

Following [36], we are interested in adapting a transformer G, while the rest (E, D, T) remain fixed. Let $\widehat{\mathsf{G}} : \mathcal{V} \times \mathcal{E} \times \Theta \to \mathcal{L}$ a modified version of a transformer G that takes learnable parameters $\boldsymbol{\theta} \in \Theta$ as an additional input. Here, $\boldsymbol{\theta}$ would represent parameters for learnable soft prompts of prompt tuning or weights of adapter tuning. Fig. 2 provides an intuitive description of $\widehat{\mathsf{G}}$ with adapter tuning.

Fine-tuning of the transformer $\widehat{\mathsf{G}}$ involves learning of newly introduced parameters $\boldsymbol{\theta}$, while existing parameters of G (*e.g.*, parameters of self-attention and cross-attention layers) remain fixed, with the learning objective as follows:

$$\boldsymbol{\theta} = \arg\min_{\boldsymbol{\theta} \in \Theta} L_{\boldsymbol{\theta}} \ , \ \ L_{\boldsymbol{\theta}} = \mathbb{E}_{(x,t) \sim \mathcal{D}_{\mathrm{tr}}, \mathbf{m} \sim \mathcal{M}} \Big[ \mathrm{CE}_{\mathbf{m}} \Big( \widehat{\mathsf{G}} \big( \mathsf{M}\left(\mathsf{E}(x), \mathbf{m}\right), \mathsf{T}(t), \boldsymbol{\theta} \big), \mathsf{E}(x) \Big) \Big], \tag{3}$$

where $\mathcal{D}_{\mathrm{tr}}$ contains a few (image, text) pairs for fine-tuning. Unlike DreamBooth [34] where the same text prompt is used to represent a set of training images, we use different text prompts for each input image to better disentangle content and style. Once trained, similarly to the procedure in Eq. (2), we synthesize images from the generation distribution of $\widehat{\mathsf{G}}(\cdot, \cdot, \boldsymbol{\theta})$. Specifically, at each decoding step $k$, we generate logits $l_k$ as follows:

$$l_k = \widehat{\mathsf{G}}\left(v_k, \mathsf{T}(t), \boldsymbol{\theta}\right) + \lambda_{\mathrm{A}}\big(\widehat{\mathsf{G}}\left(v_k, \mathsf{T}(t), \boldsymbol{\theta}\right) - \mathsf{G}\left(v_k, \mathsf{T}(t)\right)\big) + \lambda_{\mathrm{B}}\big(\mathsf{G}\left(v_k, \mathsf{T}(t)\right) - \mathsf{G}\left(v_k, \mathsf{T}(n)\right)\big), \tag{4}$$

where $\lambda_{\mathrm{A}}$ controls the level of adaptation to the target distribution by contrasting the two generation distributions, one that is fine-tuned $\widehat{\mathsf{G}}\left(v_k, \mathsf{T}(t), \boldsymbol{\theta}\right)$ and another that is not $\mathsf{G}\left(v_k, \mathsf{T}(t)\right)$, and $\lambda_{\mathrm{B}}$ controls the textual alignment by contrasting the positive ($t$) and negative ($n$) text prompts.

### 3.2.1 Constructing Text Prompts

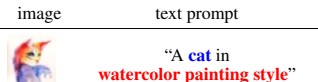

Table 1: An example text prompt at training. We construct a text prompt by composing descriptions of **content** (*e.g.*, an object) and **style** (*e.g.*, watercolor painting).

To train $\boldsymbol{\theta}$, we require training data $\mathcal{D}_{\mathrm{tr}} = \{(I_i, t_i)\}_{i=1}^N$ composed of (image, text) pairs for style reference. In many scenarios, we may be given only images as a style reference. In such cases, we need to manually append text prompts.

We propose a simple, templated approach to construct text prompts, consisting of the description of a content (*e.g.*, object, scene) followed by the phrase describing the style. For example, we use a "cat" to describe an object in Tab. 1 and append "watercolor painting" as a style descriptor. Incorporating descriptions of both content and style in the text prompt is critical, as it helps to disentangle the content from style and let learned parameters $\boldsymbol{\theta}$ model the style, which is our primary goal. While we find that using a rare token identifier [34] in place of a style descriptor (*e.g.*, "watercolor painting") works as well, having such a descriptive style descriptor provides an extra flexibility of style property editing, which will be shown in Sec. 4.4.2 and Fig. 7.

### 3.3 Iterative Training with Feedback

While our framework is generic and works well even on small training sets, the generation quality of the style-tuned model from a single image can sometimes be sub-optimal. The text construction method in Sec. 3.2.1 helps the quality, but we still find that overfitting to content is a concern. As in red boxes of Fig. 3 where the same house is rendered in the background, it is hard to perfectly avoid the content leakage. However, we see that many of the rendered images successfully disentangle style from content, as shown in the blue boxes of Fig. 3.

For such a scenario, we leverage this finding of high precision when successful and introduce an iterative training (IT) of StyleDrop using synthesized images by StyleDrop trained at an earlier stage to improve the recall (more disentanglement). We opt for a simple solution: construct a new training

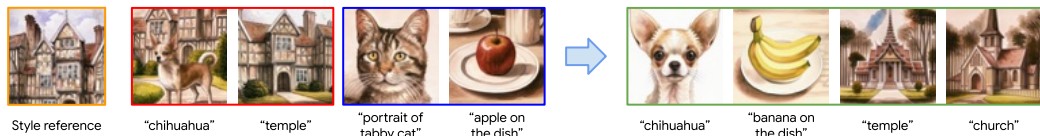

| Style reference | "chihuahua" | "temple" | "portrait of tabby cat" | "apple on the dish" | | "chihuahua" | "banana on the dish" | "temple" | "church" |

Figure 3: Iterative Training with Feedback. When trained on a single style reference image (orange box), some generated images by StyleDrop may exhibit leaked content from the style reference image (red box, images contain in the background a similar-looking house as in the style image), while other images (blue box) have better dismantlement of style from content. Iterative training of StyleDrop with the good samples (blue box) results in an overall better balance between style and text fidelity (green box).

set with a few dozen successful (image, text) pairs (*e.g.*, images in blue box of Fig. 3) while using the same objective in Eq. (3). IT results in an immediate improvement with a reduced content leakage, as in Fig. 3 green box. The key question is how to assess the quality of synthesized images.

**CLIP score** [29] measures the image-text alignment. As such, it could be used to assess the quality of generated images by measuring the CLIP score (*i.e.*, cosine similarity of visual and textual CLIP embeddings). We select images with the highest CLIP scores and we call this method an iterative training with CLIP feedback (CF). In our experiments, we find that the CLIP score to assess the quality of synthesized images is an efficient way of improving the recall (*i.e.*, textual fidelity) without losing too much style fidelity. On the other hand, CLIP score may not be perfectly aligned with the human intention [22, 40] and would not capture the subtle style property.

**Human Feedback** (HF) is a more direct way of injecting user intention into the quality evaluation of synthetic images. HF is shown to be powerful and effective in LLM fine-tuning with reinforcement learning [28]. In our case, HF could be used to compensate the CLIP score not being able to capture subtle style properties. Empirically, selecting less than a dozen images is enough for IT, and it only takes about 3 minutes per style. As shown in Sec. 4.4.4 and Fig. 9, HF is critical for some applications, such as illustration designs, where capturing subtle differences is important to correctly reflect the designer's intention. Nevertheless, due to human selection bias, style may drift or be reduced.

## 3.4 Sampling from Two $\theta$'s

There has been an extensive study on personalization of text-to-image diffusion models to synthesize images containing multiple personal assets [20, 26, 13]. In this section, we show how to combine DreamBooth and StyleDrop in a simple manner, thereby enabling personalization of both *style and content*. Inspired by the idea of diffusion as energy-based models for compositional visual generation [9, 25, 8], we sample from two modified generation distributions, guided by $\theta_s$ for style and $\theta_c$ for content, each of which are adapter parameters trained independently on style and content reference images, respectively. Unlike existing works [20, 13], our approach does not require joint training of learnable parameters on multiple concepts, leading to a greater compositional power with pre-trained adapters, which are separately trained on individual subject and style assets.

Our overall sampling procedure follows the iterative decoding of Eq. (1), with differences in how we sample logits at each decoding step. Let $t$ be the text prompt and $c$ be the text prompt without the style descriptor.[5] We compute logits at step $k$ as follows: $l_k = (1 - \gamma)l_k^s + \gamma l_k^c$, where

$$l_k^s = \widehat{\mathtt{G}}\left(v_k, \mathtt{T}(t), \theta_s\right) + \lambda_\mathrm{A}\big(\widehat{\mathtt{G}}\left(v_k, \mathtt{T}(t), \theta_s\right) - \mathtt{G}\left(v_k, \mathtt{T}(t)\right)\big) + \lambda_\mathrm{B}\big(\mathtt{G}\left(v_k, \mathtt{T}(t)\right) - \mathtt{G}\left(v_k, \mathtt{T}(n)\right)\big) \quad (5)$$

$$l_k^c = \widehat{\mathtt{G}}\left(v_k, \mathtt{T}(c), \theta_c\right) + \lambda_\mathrm{A}\big(\widehat{\mathtt{G}}\left(v_k, \mathtt{T}(c), \theta_c\right) - \mathtt{G}\left(v_k, \mathtt{T}(c)\right)\big) + \lambda_\mathrm{B}\big(\mathtt{G}\left(v_k, \mathtt{T}(c)\right) - \mathtt{G}\left(v_k, \mathtt{T}(n)\right)\big) \quad (6)$$

where $\gamma$ balances the StyleDrop and DreamBooth – if $\gamma$ is 0, we get StyleDrop, and DreamBooth if 1. By properly setting $\gamma$ (*e.g.*, $0.5 \sim 0.7$), we get images of *my content in my style* (see Fig. 5).

## 4 Experiments

We report results of StyleDrop on a variety of styles and compare with existing methods in Sec. 4.2. In Sec. 4.3 we show results on "my object in my style" combining the capability of DreamBooth and StyleDrop. Finally, we conduct an ablation study on the design choices of StyleDrop in Sec. 4.4.

## 4.1 Experimental Setting

To the best of our knowledge, there has not been an extensive study of style-tuning for text-to-image generation models. As such, we suggest a new experimental protocol.

---

[5] For example, $t$ is "A teapot in watercolor painting style" and $c$ is "A teapot".

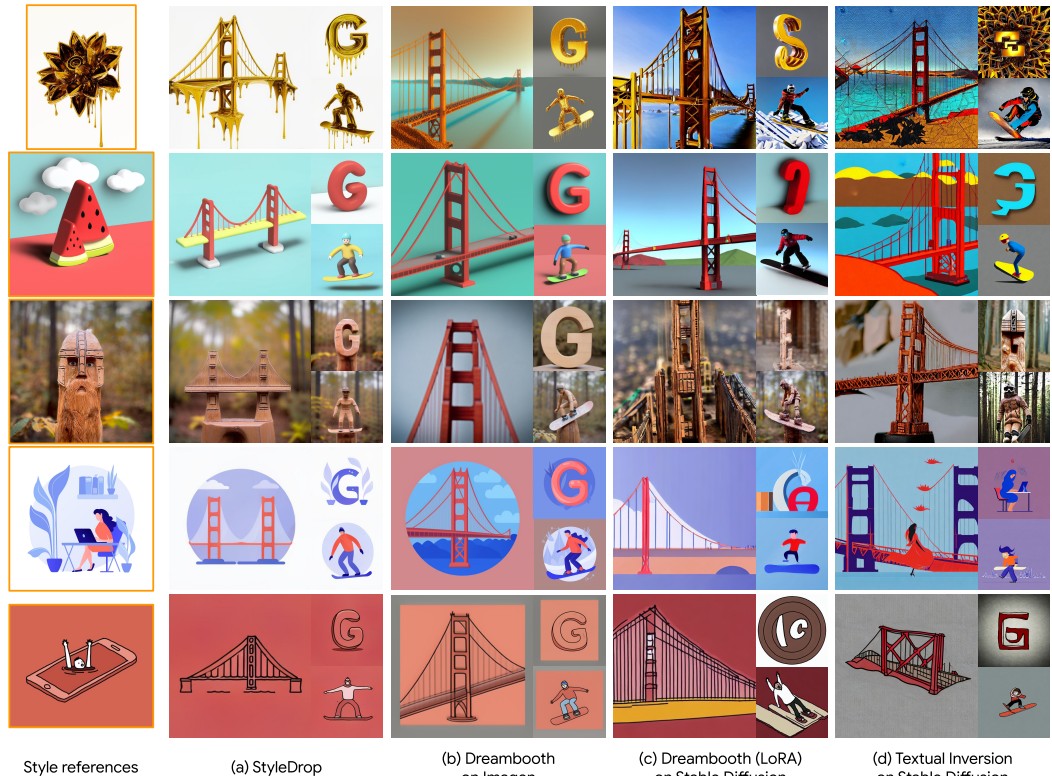

| Style references | (a) StyleDrop | (b) Dreambooth on Imagen | (c) Dreambooth (LoRA) on Stable Diffusion | (d) Textual Inversion on Stable Diffusion |

Figure 4: Qualitative comparison of style-tuned text-to-image synthesis on various styles, including "melting golden 3d rendering", "3d rendering", "wooden sculpture", "flat cartoon illustration", and "cartoon line drawing", shown on the first column. Text prompts used for synthesis are "the Golden Gate bridge", "the letter 'G'", and "a man riding a snowboard". Image sources are in Tab. S1. We see that StyleDrop (HF) consistently captures nuances such as the "melting" effect in the top row.

**Data collection.** We collect a few dozen images of various styles, from watercolor and oil painting, flat illustrations, 3d rendering to sculptures with varying materials. While painting styles have been a major focus for neural style transfer research [12, 7], we go beyond and include a more diverse set of visual styles in our experiments. We provide image sources in Tab. S1 and attribute their ownership.

**Model configuration.** As in Sec. 3.2, we base StyleDrop on Muse [5] using adapter tuning [15, 36]. For all experiments, we update adapter weights for 1000 steps using Adam optimizer [19] with a learning rate of 0.00003. Unless otherwise stated, we use "StyleDrop" to denote the second round model trained on as many as 10 synthetic images with human feedback, as in Sec. 3.3. Nevertheless, to mitigate confusion, we append "HF" (human feedback), "CF" (CLIP feedback) or "R1" (first round model) to StyleDrop whenever there needs a clarity. More training details are in Appendix B.1.

**Evaluation.** We report quantitative metrics based on CLIP [29] that measures the style consistency and textual alignment. In addition, we conduct the user preference study to assess style consistency and textual alignment. Appendix B.2 summarizes details on the human evaluation protocol.

### 4.2 StyleDrop Results

Fig. 1 shows results of our default approach on the 18 different style images that we collected, for the same text prompt. We see that StyleDrop is able to capture nuances of texture, shading, and structure across a wide range of styles, significantly better than previous approaches, enabling significantly more control over style than previously possible. Fig. 4 shows synthesized images of StyleDrop using 3 different style reference images. For comparison, we also present results of (b) DreamBooth [34] on Imagen [35], (c) a LoRA implementation of DreamBooth [34, 3, 16] and (d) textual inversion [11], both on Stable Diffusion [33].[6,7] More results are available in Figs. S9 to S14.

---

[6]Colab implementation of textual inversion [11] is used with `stable-diffusion-2`.

[7]More baselines using hard prompt made easy (PEZ) [39] are in Appendix B.

Table 2: Evaluation metrics of (top) human evaluation and (bottom) CLIP scores [29] for image-text alignment (Text) and visual style alignment (Style). We test on 6 styles from Fig. 1. For human evaluation, preferences are reported. For CLIP scores, we report the mean and standard error. We report scores for Muse [5] and Imagen [35] with styles guided by the text prompt. DB: DreamBooth, SDRP: StyleDrop, and iterative training with human feedback (HF), CLIP feedback (CF), and random selection (Random).

|  | SDRP (R1) | tie | DB on Imagen | SDRP (R1) | tie | SDRP (HF) | SDRP (HF) | tie | SDRP (CF) |
|---|---|---|---|---|---|---|---|---|---|
| Text | 31.7% | **45.0%** | 23.3% | 20.7% | **56.0%** | 23.3% | 19.4% | **58.2%** | 22.4% |
| Style | **86.0%** | 4.3% | 9.7% | **62.3%** | 7.4% | 30.3% | **60.9%** | 8.4% | 30.8% |

| Method | Imagen | DB on Imagen | Muse | StyleDrop on Muse | | | |
|---|---|---|---|---|---|---|---|
|  |  |  |  | Round 1 | HF | CF | Random |
| Text (↑) | $0.337_{\pm0.001}$ | $0.335_{\pm0.001}$ | $0.323_{\pm0.001}$ | $0.313_{\pm0.001}$ | $0.322_{\pm0.001}$ | $0.329_{\pm0.001}$ | $0.316_{\pm0.001}$ |
| Style (↑) | $0.569_{\pm0.002}$ | $0.644_{\pm0.002}$ | $0.556_{\pm0.001}$ | $\mathbf{0.705}_{\pm0.002}$ | $0.694_{\pm0.001}$ | $0.673_{\pm0.001}$ | $0.678_{\pm0.001}$ |

For baselines, we follow instructions from the respective papers and open-source implementations, but with a few modifications. For example, instead of using a rare token (*e.g.*, "a watermelon slice in [V*] style"), we use the style descriptor (*e.g.*, "a watermelon slice in 3d rendering style"), similarly to StyleDrop. We train DreamBooth on Imagen for 300 steps after performing grid-search. This is less than 1000 steps recommended in [34], but is chosen to alleviate overfitting to image content and to better capture style. For LoRA DreamBooth on Stable Diffusion, we train for 400 steps with learning rates of 0.0002 for UNet and 0.000005 for CLIP. We do not adopt the iterative training for baselines in Fig. 4. StyleDrop results without iterative training are in Sec. 4.4.3. It is clear from Fig. 4 that StyleDrop on Muse convincingly outperforms other methods that are geared towards solving subject-driven personalization of text-to-image synthesis using diffusion models.

We see that style-tuning on Stable Diffusion with LoRA DreamBooth (Fig. 4(c)) and textual inversion (Fig. 4(d)) show poor style consistency to reference images. While DreamBooth on Imagen (Fig. 4(b)) improves over those on Stable Diffusion, it still lacks the style consistency over StyleDrop on Muse across text prompts and style references. It is interesting to see such a difference as both Muse [5] and Imagen [35] are trained on the same set of image/text pairs using the same text encoder (T5-XXL [30]). We provide an ablation study to understand where the difference comes from in Sec. 4.4.1.

### 4.2.1 Quantitative Results

For quantitative evaluation, we synthesize images from a subset of Parti prompts [43]. This includes 190 text prompts of basic text compositions, while removing some categories such as abstract, arts, people or world knowledge. We test on 6 style reference images from Fig. 1.[8]

**CLIP scores.** We employ two metrics using CLIP [29], (Text) and (Style) scores. For Text score, we measure the cosine similarity between image and text embeddings. For Style score, we measure the cosine similarity between embeddings of style reference and synthesized images. We generate 8 images per prompt for 190 text prompts, 1520 images in total. While we desire high scores, these metrics are not perfect. For example, Style score can easily get to 1.0 if mode collapses.

StyleDrop results in competitive Text scores to Muse (*e.g.*, 0.323 vs 0.322 of StyleDrop (HF)) while achieving significantly higher Style scores (*e.g.*, 0.556 vs 0.694 of StyleDrop (HF)), implying that synthesized images by StyleDrop are consistent in style with style reference images, without losing text-to-image generation capability. For the 6 styles we test, we see a light mode collapse from the first round of StyleDrop, resulting in a slightly reduced Text score. Iterative training (IT) improves the Text score, which is aligned with our motivation. As a trade-off, however, they show reduced Style scores over Round 1 models, as they are trained on synthetic images and styles may have been drifted due to a selection bias.

DreamBooth on Imagen falls short of StyleDrop in Style score (0.644 vs 0.694 of HF). We note that the increment in Style score for DreamBooth on Imagen is less significant ($0.569 \rightarrow 0.644$) than StyleDrop on Muse ($0.556 \rightarrow 0.694$). We think that the fine-tuning for style on Muse is more effective than that on Imagen. We revisit this in Sec. 4.4.1.

**Human evaluation.** We formulate 3 binary comparison tasks for user preference among StyleDrop (R1), StyleDrop with different feedback signals, and DreamBooth on Imagen. Users are asked to select their preferred result in terms of style and text fidelity between images generated from two

---

[8]Images used are (1, 2), (1, 6), (2, 3), (3, 1), (3, 5), (3, 6) of Fig. 1, and are also visualized in Fig. S8.

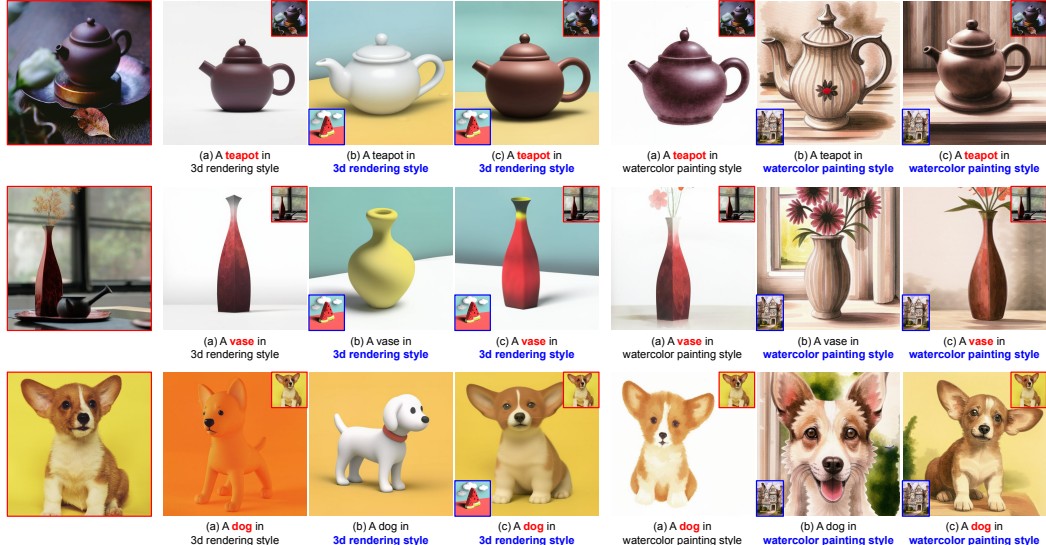

Figure 5: Qualitative comparison of (a) DreamBooth, (b) StyleDrop, and (c) DreamBooth + StyleDrop. For DreamBooth and StyleDrop, style and subject are guided by text prompts, respectively, whereas DreamBooth + StyleDrop, both style (blue inset box at bottom left) and subject (red inset box at top right) are guided by respective reference images. Image sources are in Tab. S1.

different models (*i.e.*, an A/B test), while given a style reference image and the text prompt. Details on the study is in Appendix B.2. Results are in Tab. 2 (top). Compared to DreamBooth on Imagen, images by StyleDrop are significantly more preferred by users in `Style` score. The user study also shows style drifting more clearly when comparing StyleDrop (R1) and StyleDrop IT either by HF or CF. Between HF and CF, HF retains better `Style` and CLIP retained better `Text`. Overall, we find that CLIP scores are a good proxy to the user study.

## 4.3 My Object in My Style

We show in Fig. 5 synthesized images by sampling from two personalized generation distributions, one for an object and another for the style, as described in Sec. 3.4. To learn object adapters, we use 5∼6 images per object.[9] Style adapters from Sec. 4.2 are used without any modification. The value of $\gamma$ (to balance the contribution of object and style adapters) is chosen in the range 0.5–0.7. We show synthesized images from (a) object adapter only (*i.e.*, DreamBooth), (b) style adapter only (*i.e.*, StyleDrop), and (c) both object and style adapters. We see from Fig. 5(a) that text prompts are not sufficient to generate images with styles we desire. From Fig. 5(b), though StyleDrop gets style correct, it generates objects that are inconsistent with reference subjects. The proposed sampling method from two distributions successfully captures both *my object* and *my style*, as in Fig. 5(c).

## 4.4 Ablations

We conduct ablations to better understand StyleDrop. In Sec. 4.4.1 we compare the behavior of the Imagen and Muse models. In Sec. 4.4.2 we highlight the importance of a style descriptor. In Sec. 4.4.3, we compare choices of feedback signals for iterative training. In Sec. 4.4.4, we show to what extent StyleDrop learns distinctive styles properties from reference images.

### 4.4.1 Comparative Study of DreamBooth on Imagen and StyleDrop on Muse

We see in Sec. 4.2 that StyleDrop on Muse convincingly outperforms DreamBooth on Imagen. To better understand where the difference comes from, we conduct some control experiments.

**Impact of training text prompt.** We note that both experiments in Sec. 4.2 are carried out using the proposed descriptive style descriptors. To understand the contribution of *fine-tuning*, rather than the prompt engineering, we conduct control experiments, one with a rare token as in [34] (*i.e.*, "A flower in [V*] style") and another with descriptive style prompt.

---

[9]`teapot` and `vase` images from DreamBooth [34] are used.

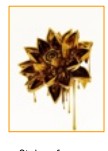 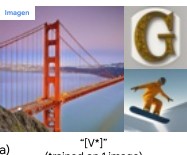 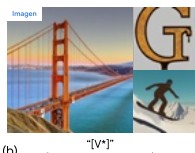 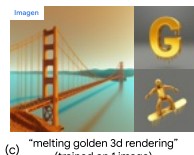 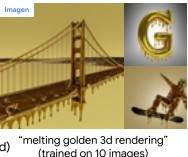

| | | | |
|---|---|---|---|
| (a) "[V*]" (trained on 1 image) | (b) "[V*]" (trained on 10 images) | (c) "melting golden 3d rendering" (trained on 1 image) | (d) "melting golden 3d rendering" (trained on 10 images) |

Figure 6: Ablation study using Imagen [35]. (a, b) are trained with a rare token and (c, d) are trained with a style descriptor. (a, c) are trained on a single style reference image. (b, d) are trained on 10 synthetic images from StyleDrop. With Imagen, we need 10 images and a descriptive style descriptor to capture the style, as in (d).

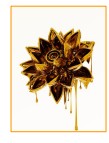 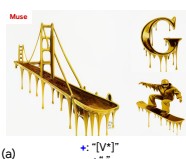 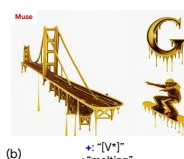 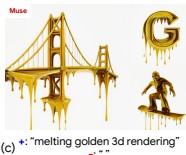 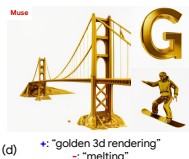

Figure 7: StyleDrop on Muse. All models are trained on 10 images from StyleDrop, which in turn was trained on a single style image. (a, b) are trained with a rare token and (c, d) are trained with a style descriptor. When trained with a descriptive style descriptor, StyleDrop can support additional applications such as *style editing* (d), here removing the "melting" component of the reference style.

Results on Muse are in Fig. 7. Comparing (a) and (c), we do not find a substantial change, suggesting that the style can be *learned* via an adapter tuning without too much help of a text prior. On the other hand, as seen in Fig. 6, comparing (a) and (c) with Imagen as a backbone model, we see a notable difference. For example, "melting" property only appears for some images synthesized from a model trained with the descriptive style descriptor. This suggests that the learning capability of fine-tuning on Imagen may not be as powerful as that of Muse, when given only a few training images.

**Data efficiency.** Next, we study whether the quality of the fine-tuning on Imagen could be improved with more training data. In this study, we train a DreamBooth on Imagen using 10 human selected, synthetic images from StyleDrop on Muse. Results are in Fig. 6. Two models are trained with (b) a rare token and (d) a descriptive style descriptor. We see that the style consistency improves a lot when comparing (c) and (d) of Fig. 6, both in terms melting and golden properties. However, when using a rare token, we do not see any notable improvement from (a) to (b). This suggests that the superiority of StyleDrop on Muse may be coming from its extraordinary fine-tuning data efficiency.

### 4.4.2 Style Property Edit with Concept Disentanglement

We show in Sec. 4.4.1 that StyleDrop on Muse is able to learn the style using a rare token identifier. Then, what is the benefit of descriptive style descriptor? We argue that not all styles are described in a single word and the user may want to learn style properties selectively. For example, the style of an image in Fig. 7 may be written as a composite of "melting", "golden", and "3d rendering", but the user may want to learn its "golden 3d rendering" style without "melting".

We show that such a *style property edit* can be naturally done with a descriptive style descriptor. As in Fig. 7(d), learning with a descriptive style descriptor provides an extra knob to edit a style by omitting certain words (*e.g.*, "melting") from the style descriptor at synthesis. This clearly shows the benefit of descriptive style descriptors in disentangling visual concepts and creating a new style based on an existing one. This is less amenable when trained with the rare token, as in Fig. 7(b).

### 4.4.3 Iterative Training with Different Feedback Signals

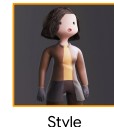 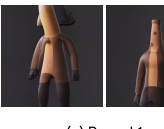 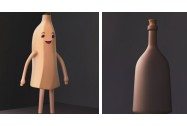 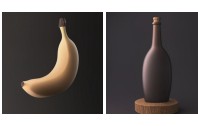 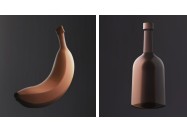

| | | | |
|---|---|---|---|
| Style reference | (a) Round 1 | (b) Random | (c) CLIP | (d) Human |

Figure 8: Qualitative comparison of StyleDrop. (a) Round 1, (b) IT with random selection, (c) CLIP and (d) Human feedback. Generated images of "a banana" and "a bottle" in "3d rendering style" are visualized. While StyleDrop Round 1 model captures the style very well, it often suffer from a content leakage (*e.g.*, a banana and women are mixed). (c, d) IT with a careful selection of synthetic images reduces content leakage and improves.

We study how different feedback signals affects the performance of StyleDrop. We compare three feedback signals, including human, CLIP, and random. For CLIP and random signals, we synthesize

64 images per prompt from 30 text prompts and select one image per prompt. For human, we select 10 images from the same pool, which takes about 3 minutes per style. See Appendix B.2 for details.

Qualitative results are in Fig. 8. We observe that some images in (a) from a Round 1 model show a mix of banana or bottle with a human. Such concept leakage is alleviated with IT, though we still see a banana with arms and legs with Random strategy. The reduction in concept leakage could be verified with the `Text` score, achieving (a) 0.303, (b) 0.322, (c) 0.339, and (d) 0.328. On the other hand, `Style` score, (a) 0.560, (b) 0.567, (c) 0.542, and (d) 0.549, could be misleading in this case, as we compute the visual similarity to the style reference image, favoring a content leakage. Between CLIP and human feedback, we see a clear trade-off between text and style fidelity from quantitative metrics.

#### 4.4.4 Fine-Grained Style Control with User Intention

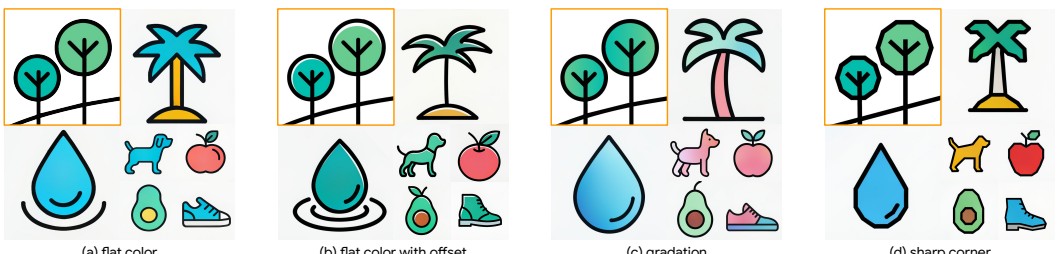

(a) flat color     (b) flat color with offset     (c) gradation     (d) sharp corner

Figure 9: Fine-grained style control. StyleDrop captures subtle style differences, such as (b) color offset, (c) gradation, or (d) sharp corner, reflecting designer's intention in text-to-image synthesis.

Moreover, human feedback is more critical when trying to capture subtle style properties. In this study, we conduct experiments on four images in Fig. 9 inside orange boxes, created by the same designer with varying style properties, such as color offset (Fig. 9(b)), gradation (Fig. 9(c)), and sharp corners (Fig. 9(d)). We train two more rounds of StyleDrop with human feedback. We use the same style descriptor of "minimal flat illustration style" to make sure the same text prior is given to all experiments. As in Fig. 9, style properties such as color offset, gradation, and corner shape are captured correctly. This suggests that StyleDrop offers the control of fine-grained style variations.

## 5   Conclusion

We have presented StyleDrop, a novel approach to enable synthesis of any style through the use of a few user-provided images of that style and a text description. Built on Muse [5] using adapter tuning [15], StyleDrop achieves remarkable style consistency at text-to-image synthesis. Training StyleDrop is efficient both in the number of learnable parameters (*e.g.*, < 1%) and the number of style samples (*e.g.*, 1) required.

**Limitations.** Visual styles are of course even more diverse than what is possible to explore in our paper. More study with a well-defined system of visual styles, including, but not limited to, the formal attributes (*e.g.*, use of color, composition, shading), media (*e.g.*, line drawing, etching, oil painting), history and era (*e.g.*, Renaissance painting, medieval mosaics, Art Deco), and style of art (*e.g.*, Cubism, Minimalism, Pop Art), would broaden the scope. While we show in part the superiority of a generative vision transformer to diffusion models at few-shot transfer learning, it is by no means conclusive. We leave an in-depth study among text-to-image generation models as a future work.

**Societal impact.** As illustrated in Fig. 4, StyleDrop could be used to improve the productivity and creativity of art directors and graphic designers when generating various visual assets in their own style. StyleDrop makes it easy to reproduce many personalized visual assets from as little as one seed image. We recognize potential pitfalls such as the ability to copy individual artists' styles without their consent, and urge the responsible use of our technology.

**Acknowledgement.** We thank Varun Jampani, Jason Baldridge, Forrester Cole, José Lezama, Steven Hickson, Kfir Aberman for their valuable feedback on our manuscript.

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

# A Image Attributions

We provide the links to images used for style references (please click through on each number).

Table S1: Image sources.

| | |
|---|---|
| Fig. 1 | (row 1) 1, 2, 3, 4, 5, 6; (row 2) 1, 2, 3, 4, 5, 6; (row 3) 1, 2, 3, 4, 5, 6 |
| Figs. S6 and S7 | (row 1) 1, 2, 3, 4, 5, 6, 7, 8; (row 2) 1, 2, 3, 4, 5, 6, 7, 8; (row 3) 1, 2, 3, 4, 5, 6, 7, 8 |
| Fig. 9 | Special thanks to Elizabeth Cruz for help designing the initial style image. |

Table S2: A list of text prompts used to synthesize images for additional round of training of StyleDrop. {}'s are filled with the style descriptors, *e.g.*, "in watercolor painting style".

| | | |
|---|---|---|
| "A chihuahua {}" | "A chihuahua walking on the street {}" | "A chihuahua walking in the forest {}" |
| "A tabby cat {}" | "A tabby cat walking on the street {}" | "A tabby cat walking in the forest {}" |
| "A portrait of chihuahua {}" | "A portrait of tabby cat {}" | "A portrait of human face {}" |
| "An apple on the table {}" | "An apple on the dish {}" | "An apple on the ground {}" |
| "A banana on the table {}" | "A banana on the dish {}" | "A banana on the ground {}" |
| "A human {}" | "A human walking on the street {}" | "A human walking in the forest {}" |
| "A church on the street {}" | "A temple on the street {}" | "A cabin on the street {}" |
| "A church in the mountain {}" | "A temple in the mountain {}" | "A cabin in the mountain {}" |
| "A church in the field {}" | "A temple in the field {}" | "A cabin in the field {}" |
| "A church on the beach {}" | "A temple on the beach {}" | "A cabin on the beach {}" |

# B Experiments

## B.1 Details on Model Training

### B.1.1 Adapter Architecture

We apply an adapter at every layer of transformer. Specifically, following [15], we apply two adapters for each layer, one after the cross-attention block, and another after the MLP block. An example code explaining how to apply an adapter to the output of an attention layer and how to generate adapter weights are in Fig. S1. All up weights (wu) are initialized with zeros, and down weights (wd) are initialized from truncated normal distribution with standard deviation of 0.02.

We note that adapter weights are generated in a parameter-efficient way via weight sharing across transformer layers. This is triggered by setting is_shared to True, and the total number of parameters would be reduced roughly by the number of transformer layers. The number of parameters of adapter weights are given in Tab. S3. While we use these settings on all experiments, one can easily reduce the number of parameters by setting is_shared to True for Base (Round 2) and Super-res fine-tuning without loss in quality.

### B.1.2 Hyperparameters

We provide in Tab. S3 hyperparameters for optimizer, adapter architecture, and synthesis. Note that we use the batch size of 8, 1 per core of TPU v3, but StyleDrop can be also optimized on a single GPU (*e.g.*, A100) with batch size of 1. We find that learning rate higher than 0.00003 for the base model often results in overfitting to content of a style reference image. Learning rate lower than 0.00003 for the base model leads to slower convergence and we suggest to increase the number of train steps in such a case.

### B.1.3 Style Descriptors

We provide full description on descriptive style descriptors for images used in our experiments in Tab. S4. As discussed in Sec. 4.4.1 and Sec. 4.4.2, StyleDrop works well without descriptive style descriptors, but they add additional capability such as style editing.

```
1   import flax.linen as nn
2   import jax
3   import jax.numpy as jnp
4
5   def apply_adapter(emb, wd, wu):
6     """Applies adapter.
7
8     Args:
9       emb: token embedding, B x S x D.
10      wd: down weight, D x H.
11      wu: up weight, H x D.
12
13    Returns:
14      tensor, B x S x D.
15    """
16
17    prj = jnp.einsum('...d,dh->...h', emb, wd)
18    prj = jax.nn.gelu(prj)
19    prj = jnp.einsum('...h,hd->...d', prj, wu)
20    return emb + prj
21
22
23  class AdapterGenerator(nn.Module):
24    """Generates Adapter Weights."""
25
26    d_emb: int  # Embedding dimension.
27    d_prj: int  # Projection dimension.
28    n_layer: int  # Number of transformer layers.
29    is_shared: bool  # Share adapter parameters across layers.
30
31    @nn.compact
32    def __call__(self):
33      D, H, L = self.d_emb, self.d_prj, self.n_layer
34      idx = jnp.arange(L)
35      if self.is_shared:
36        idx0 = jnp.zeros_like(idx)
37        # Factorize depth, emb and prj.
38        dd = nn.Embed(L, H)(idx).reshape(L, 1, H)
39        du = nn.Embed(L, D)(idx).reshape(L, 1, D)
40        wd = nn.Embed(1, D * H)(idx0).reshape(L, D, H) + dd
41        wu = nn.Embed(1, H * D)(idx0).reshape(L, H, D) + du
42      else:
43        wd = nn.Embed(L, D * H)(idx).reshape(L, D, H)
44        wu = nn.Embed(L, H * D)(idx).reshape(L, H, D)
45      return wd, wu
```

Figure S1: An example code on how to apply an adapter and how adapter weights are generated in Flax-ish format.

## B.2  Details on Human Evaluation

In this section, we provide more details on the user preference study discussed in Sec. 4.2.1. 3 binary comparison tasks are conducted between DreamBooth on Imagen and StyleDrop (Round 1), StyleDrop (Round 1) and StyleDrop (IT human), StyleDrop (IT human) and StyleDrop (IT CLIP). 300 queries (50 queries per style from 6 styles, as shown in Fig. S8 are uploaded. The same query is asked to 5 raters independently to mitigate the human selection bias and variance. In total, we collect 4500 answers.

We show in Fig. S2 the screenshot of the interface. We provide an instruction, examples, and task, composed of a reference image, text prompt, and two images that raters are asked to compare.

**Instructions.**

Table S3: Hyperparameters for optimizer, adapter architecture, and synthesis.

|  | Base (Round 1) | Base (Round 2) | Super-res |
|---|---|---|---|
| Learning rate | 0.00003 | 0.00003 | 0.0001 |
| Batch size | 8 | 8 | 8 |
| # steps | 1000 | 1000 | 1000 |
| `d_prj` | 4 | 32 | 32 |
| `is_shared` | True | False | False |
| # adapter parameters | 0.23M | 12.6M | 6.3M |
| # decoding step | 36 | 36 | 12 |
| temperature | 4.5 | 4.5 | 4.5 |
| $\lambda_A$ | 0.0–2.0 | 2.0 | 1.0 |
| $\lambda_B$ | 5.0 | 5.0 | 0.0 |

- Task: Given a reference image and two machine-generated output images, select which machine-generated output better matches the style of the reference image.

- Review this definition of a style: style (from Merriam-Webster): A particular manner or technique by which something is done, created or performed. Then choose either Image A, Image B, or Cannot Determine / Both Equally.

- Next, review the reference text. Select which generated output is best described by the reference text. If you're again not sure, select Cannot Determine / Both Equally.

**Questions.**

- Which Machine-Generated Image best matches the style of the reference image? Image A, Image B, Cannot Determine / Both Equally

- Which Machine-Generated Image is best described by the reference text? Image A, Image B, Cannot Determine / Both Equally

**Additional Analysis.** We provide an additional analysis on the user preference study results. Note that the numbers reported in Tab. 2 are based on the majority voting and claimed tie only if two models received the same number of votes. To provide a full picture, we draw a diagram that shows individual vote counts in Fig. S3. We find that there are more "tie" counts, but confirm overall a consistent trend with the results by the majority vote in Tab. 2.

### B.3   Extended Ablation Study

#### B.3.1   Classifier-Free Guidance

We conduct an ablation study on classifier-free guidance (CFG) parameters, $\lambda_A$ and $\lambda_B$, of Eq. (4). They play different roles: $\lambda_A$ controls the level of style adaptation and $\lambda_B$ controls the text prompt fidelity. We conduct two sets of experiments, one with the StyleDrop Round 1 model and another with the model trained with a human feedback (IT, human).

**StyleDrop (Round 1) model.** In this study, we use StyleDrop (Round 1) model, which is trained on a single style reference image.

1. **$\lambda_A$ with fixed $\lambda_B$.** Firstly, we vary $\lambda_A$ while fixing $\lambda_B$ to 5.0. Results are in Fig. S4a. When $\lambda_A = 0.0$, we find synthesized images having less faithful styles to the style reference images. As we increase $\lambda_A$, we see the style of synthesized images getting more consistent. However, when $\lambda_A$ becomes too large (*e.g.*, 5.0) the style factor dominates the sampling process, making the content of synthesized images collapsed to that of the style reference image and hard to make it follow the text condition.

2. **$\lambda_B$ with fixed $\lambda_A$.** Subsequently, we investigate the impact of $\lambda_B$, while fixing $\lambda_A = 2.0$. Results are in Fig. S4c. When $\lambda_B = 0.0$, we see that synthesized images being collapsed to

Table S4: Text prompts used for experiments in Fig. 1. We construct a text prompt by composing descriptions of a content (*e.g.*, object) and style (*e.g.*, watercolor painting).

| image | text prompt | image | text prompt |
|-------|-------------|-------|-------------|
|  | "A house in watercolor painting style" |  | "A cat in watercolor painting style" |
|  | "Flowers in watercolor painting style" |  | "A village in oil painting style" |
|  | "A village in line drawing style" |  | "A portrait of a person wearing a hat in oil painting style" |
|  | "A person drowning into the phone in cartoon line drawing style" |  | "A woman walking a dog in flat cartoon illustration style" |
|  | "A woman working on a laptop in flat cartoon illustration style" |  | "A Christmas tree in sticker style" |
|  | "A wave in abstract rainbow colored flowing smoke wave design" |  | "A mushroom in glowing style" |
|  | "Slices of watermelon and clouds in the background in 3d rendering style" |  | "A thumbs up in glowing 3d rendering style" |
|  | "A woman in 3d rendering style" |  | "A bear in kid crayon drawing style" |
|  | "A flower in melting golden 3d rendering style" |  | "A Viking face with beard in wooden sculpture" |

the style reference image without being too much text controlled. Increasing $\lambda_B$ improves the text fidelity, but eventually override the learned style to a more generic, text-guided style of Muse model.

Overall, we verify that $\lambda_A$ and $\lambda_B$ play roles as intended to control the style adaptation and the text prompt conditioning. Nonetheless, these two parameters impact each other and we find that fixing $\lambda_B = 5.0$ and control $\lambda_A$ to trade-off the style and text fidelity is sufficient for most cases.

**StyleDrop IT (human) model.** Next, we use StyleDrop (IT, human), which is trained on synthetic images manually selected with a human feedback. We find that the model becomes less sensitive to the change of guidance scales $\lambda_A$ or $\lambda_B$ and more robust across values in terms of content disentanglement and style consistency.

### B.3.2   Visual Comparison to Muse Baseline

In addition to the Tab. 2, we provide a comparison of StyleDrop to the Muse baseline, *i.e.*, generating stylized images through only a text prompting. We show results in Fig. S5. As we see, images generated by Muse are in more generic styles, while images by StyleDrop follow the style of reference images more closely.

**Instructions**

**Task:** Given a reference image and two machine-generated output images, select which machine-generated output better matches the **style** of the reference image.

Review this definition of a style:

**style** (from Merriam-Webster): *A particular manner or technique by which something is done, created or performed.*

Then choose either Image A, Image B, or **Cannot Determine / Both Equally.**

Next, review the reference text. Select which generated output is best described by the reference text.

If you're again not sure, select **Cannot Determine / Both Equally.**

**View Some Examples:**

Example A: Given a watercolor painting of a scene, the style is the use of watercolors, the specific color palette, the brush strokes, etc. You are assessing which of the outputs best matches the watercolor style of the reference image.

Example B: Given a black and white pencil drawing done using only dots (see: Black and White Pointilism), the style is the use of a pencil and the use of only dots to portray the subject / image. Choose the generated image that best matches that technique.

Example C: Given a 3D rendered image of an octopus, the style is the use of a particular type of 3D rendering. Choose the generated image that best matches that type of 3D rendering.

For the reference text, choose which generated output is best described by the text.

For example, if the reference text is **An octopus in a watercolor style** , choose which example more accurately displays an octopus in a watercolor style, without regard for the reference image.

**Task**

**Reference Image**

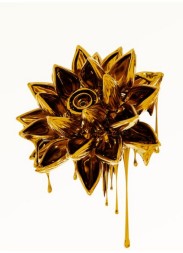

| Machine-Generated Image A | Machine-Generated Image B | Selection |
|---|---|---|
| | | Which Machine-Generated Image best matches the style of the reference image?
○ Image A
○ Image B
○ Cannot Determine / Both Equally

**Reference Text:** A dolphin, animals, in melting gold 3d rendering style
Which Machine-Generated Image is best described by the reference text?
○ Image A
○ Image B
○ Cannot Determine / Both Equally
Submit |

Figure S2: User preference study interface. We cast the problem as a binary comparison task and ask raters to choose one from two images that is best aligned with the style reference image or the text prompt. The same query is asked to 5 different raters, with location of two images randomized.

### B.3.3 Quantitative Results on Descriptive Text Prompts

In this section, we ablate the importance of descriptive text prompts. To this end, we compare two one-shot fine-tuning methods on Muse, first, using a descriptive text prompt (a.k.a. StyleDrop, as in Sec. 3.2.1), and second, using a rare token (a.k.a. DreamBooth), for style descriptors. We measure the `Text` and `Style` CLIP scores as in Tab. 2. We observe higher scores on both `Text` (0.313 vs 0.308) and `Style` (0.705 vs 0.654) with StyleDrop on Muse than DreamBooth on Muse.

### B.3.4 Robustness of Human Feedback

StyleDrop involves an iterative training with human feedback and the performance may vary across different users. In this section, we conduct experiments of iterative training based on feedback from 5 different human subjects and see the robustness of our method. Experiments are done on 6 styles considered in Tab. 2. We obtain `Style` score of 0.687 on average with the standard deviation of 0.011 and `Text` score of 0.325 on average with the standard deviation of 0.003. We find that there is a clear variance based on the human subject and their quality of feedback, but the standard deviation is low.

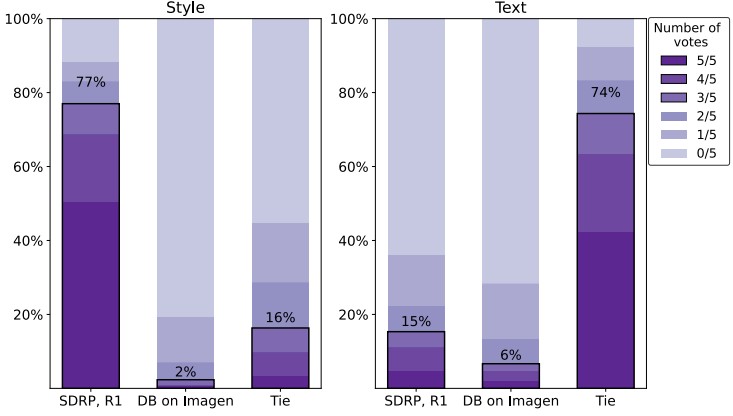

(a) StyleDrop Round 1 (SDRP, R1) vs. DreamBooth on Imagen.

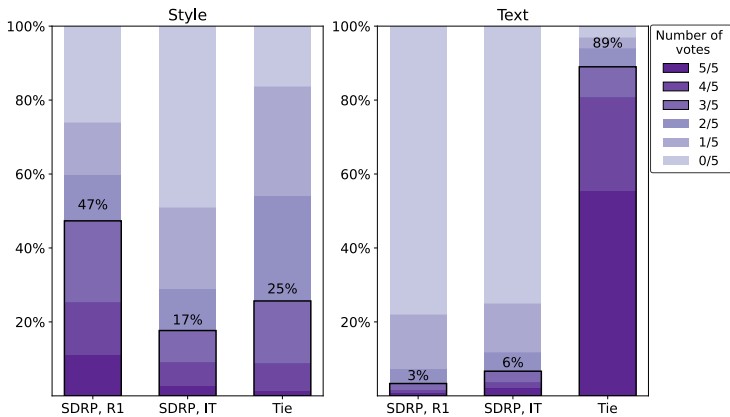

(b) StyleDrop Round 1 (SDRP, R1) vs. StyleDrop IT (human) (SDRP, IT).

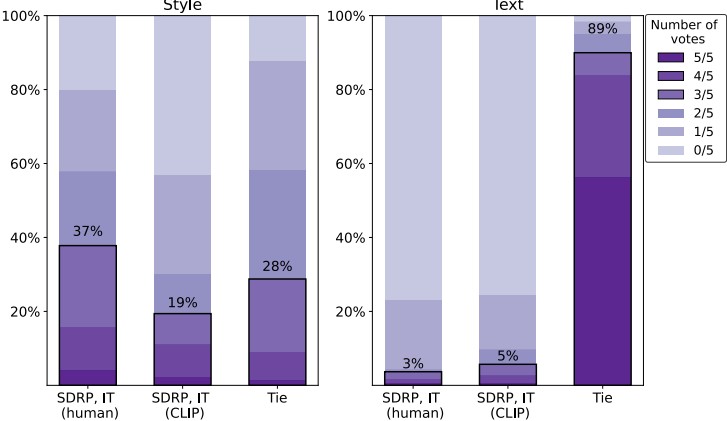

(c) StyleDrop IT (human) (SDRP, IT (human)) vs. StyleDrop IT (CLIP) (SDRP, IT (CLIP)).

Figure S3: Comprehensive analysis on user preference study.

## B.4    Extended Baseline Comparison

In addition to Sec. 4.2 and Fig. 4, we provide additional qualitative comparison with hard prompt made easy (PEZ) [39] in Fig. S15. We do not find PEZ to be better than any of the compared methods including StyleDrop and DreamBooth.

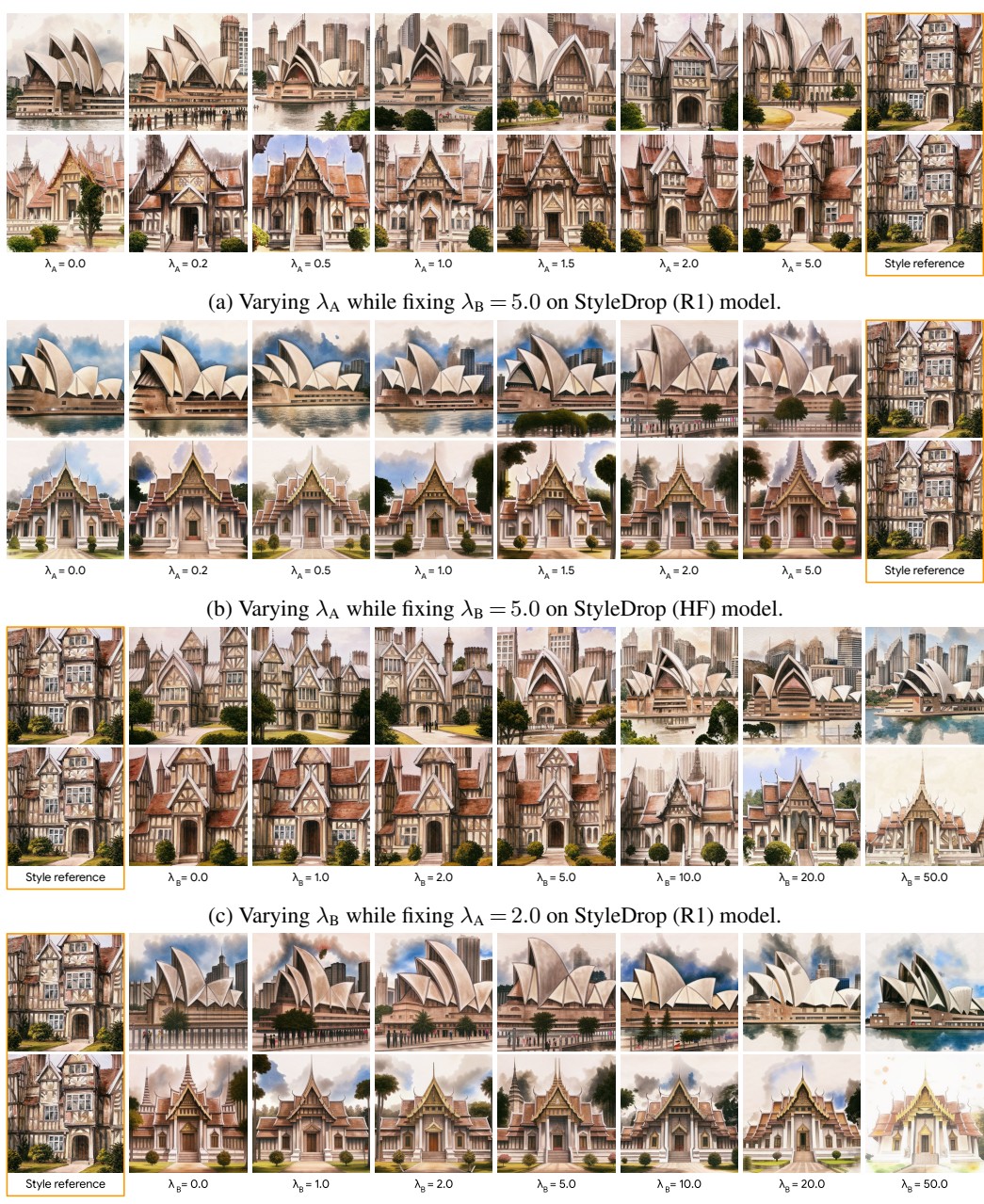

(a) Varying $\lambda_A$ while fixing $\lambda_B = 5.0$ on StyleDrop (R1) model.

(b) Varying $\lambda_A$ while fixing $\lambda_B = 5.0$ on StyleDrop (HF) model.

(c) Varying $\lambda_B$ while fixing $\lambda_A = 2.0$ on StyleDrop (R1) model.

(d) Varying $\lambda_B$ while fixing $\lambda_A = 2.0$ on StyleDrop (HF) model.

Figure S4: Ablation study on the classifier-free guidance (CFG) scales $\lambda_A$ and $\lambda_B$ of Eq. (4) on (a, c) StyleDrop (R1) and (b, d) StyleDrop (HF). $\lambda_A$ controls the style adaptation and $\lambda_B$ controls the text prompt condition. StyleDrop (R1) model responds to CFG scales sensibly and shows issues such as a content leakage (*e.g.*, with large values of $\lambda_A$, or small values of $\lambda_B$). On the other hand, StyleDrop (HF) model shows robustness to the change of guidance scales. Text prompts are "An Opera house in Sydney" and "A temple".

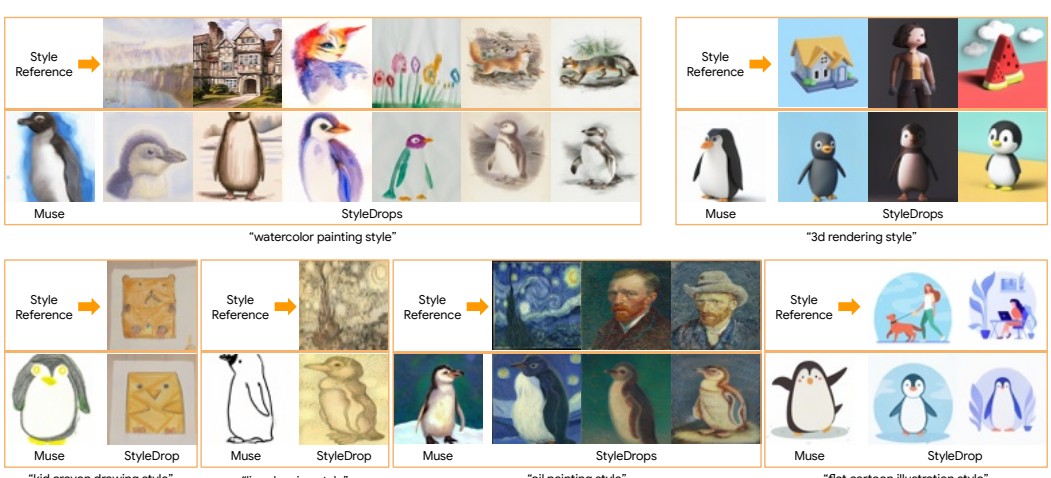

Figure S5: Comparison of generated images by Muse and StyleDrop. Muse generates images only through the text prompt, whereas StyleDrop generates images by text prompt and the style tuning using style reference images. For each orange box, we show style reference images in the first row, and generated images in the second row by Muse (first column) and StyleDrop. Images generated by StyleDrop follows the style of the reference images, while images generated by Muse represent more generic styles.

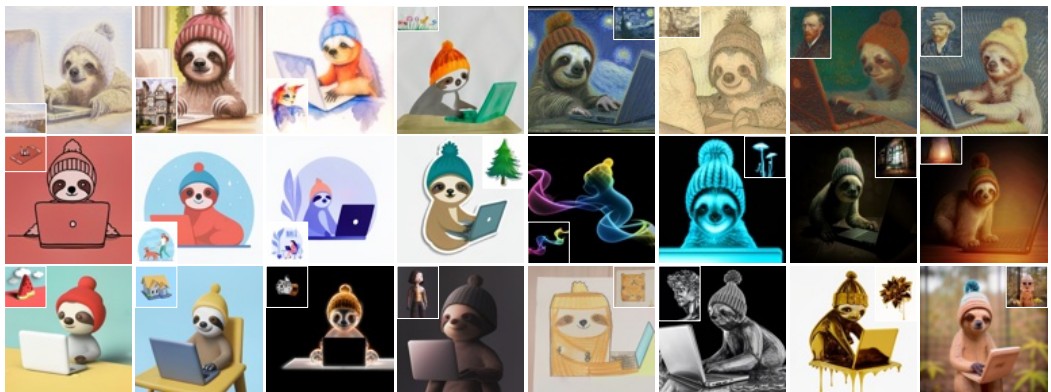

(a) "*A fluffy baby sloth with a knitted hat trying to figure out a laptop, close up*"

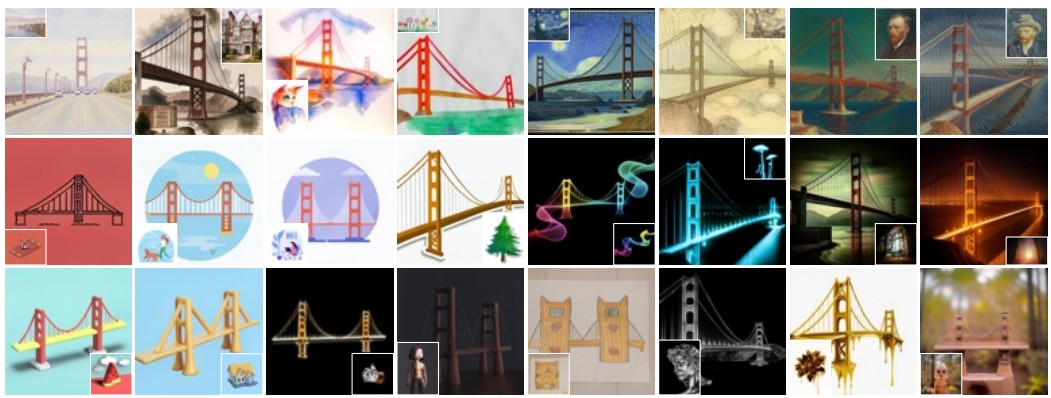

(b) "*A Golden Gate bridge*"

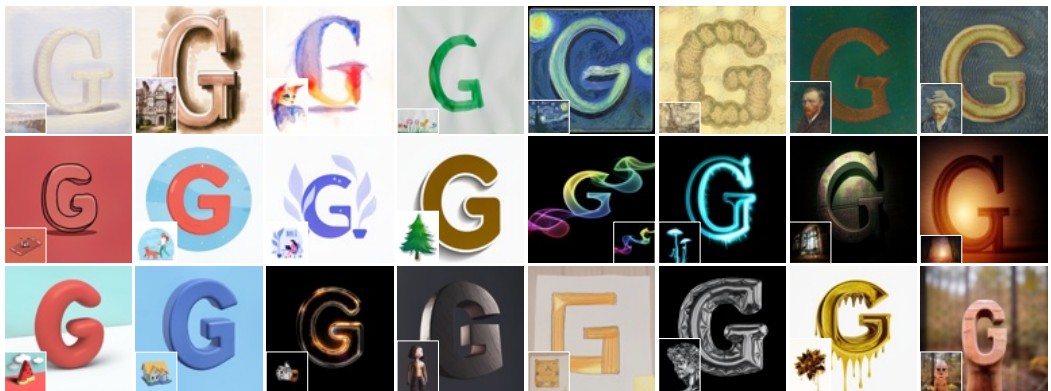

(c) "*The letter 'G'*"

Figure S6: StyleDrop on 24 styles. Style descriptors are appended to each text prompt.

(a) "*A man riding a snowboard*"

(b) "*A panda eating bamboo*"

(c) "*A friendly robot*"

Figure S7: StyleDrop on 24 styles. Style descriptors are appended to each text prompt.

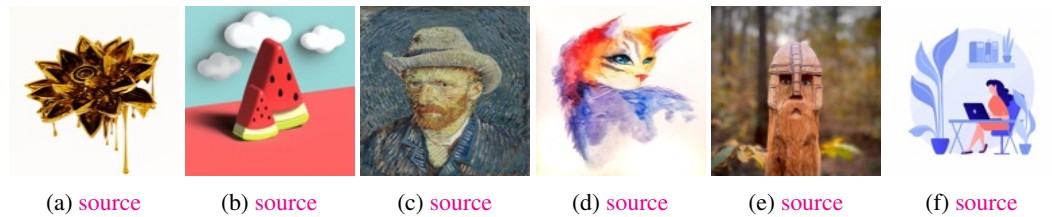

(a) source     (b) source     (c) source     (d) source     (e) source     (f) source

Figure S8: Style reference images. (a) "melting golden 3d rendering", (b) "3d rendering", (c) "oil painting", (d) "watercolor painting", (e) "wooden sculpture", (f) "flat cartoon illustration".

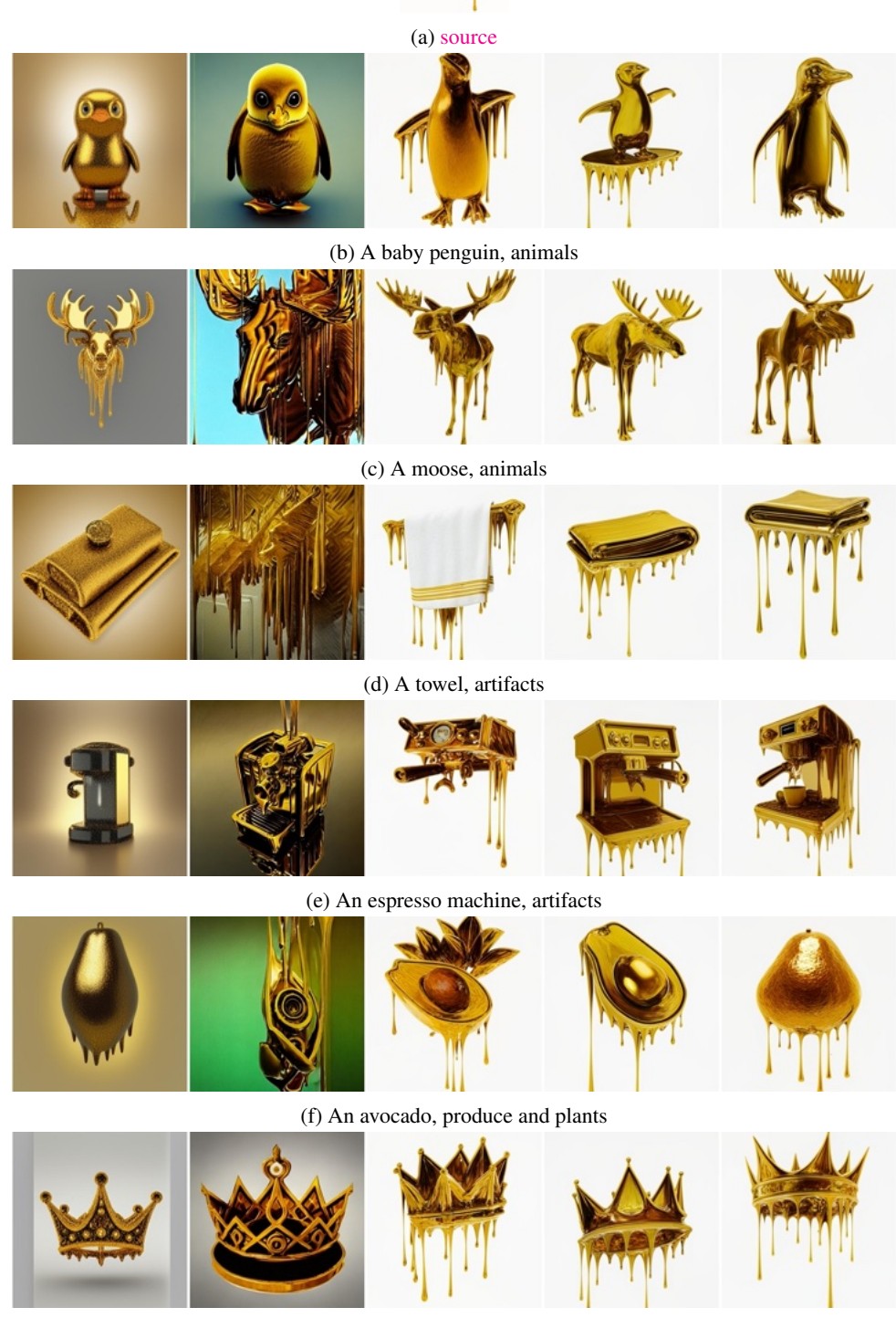

(a) source

(b) A baby penguin, animals

(c) A moose, animals

(d) A towel, artifacts

(e) An espresso machine, artifacts

(f) An avocado, produce and plants

(g) A crown, artifacts

Figure S9: Results comparison without cherry-picking. From left to right, DreamBooth on Imagen, LoRA DreamBooth on Stable Diffusion, StyleDrop (round 1), StyleDrop (round 2, HF), StyleDrop (round 2, CLIP score feedback). A style reference image is shown in Fig. S8a.

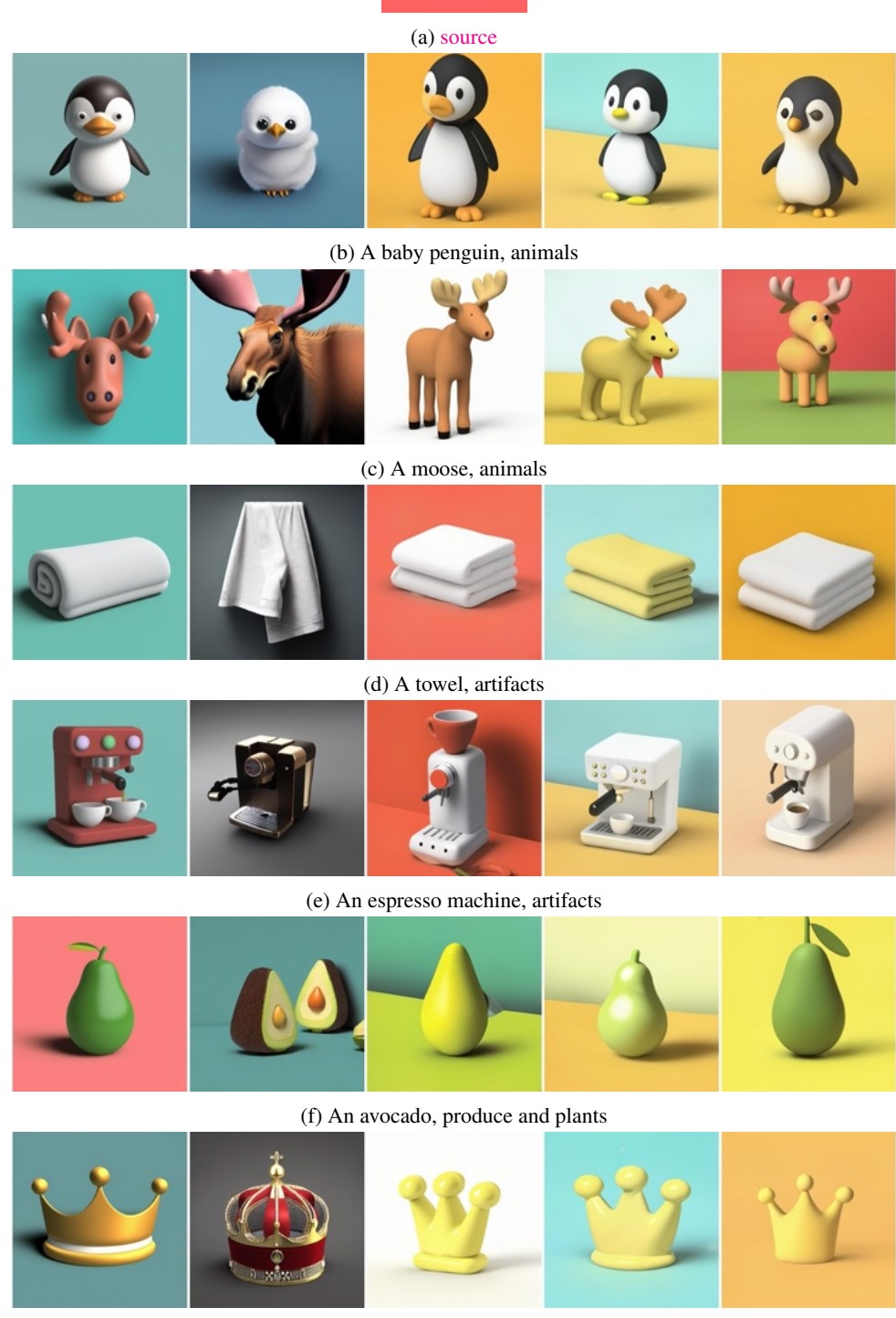

(a) source

(b) A baby penguin, animals

(c) A moose, animals

(d) A towel, artifacts

(e) An espresso machine, artifacts

(f) An avocado, produce and plants

(g) A crown, artifacts

Figure S10: Results comparison without cherry-picking. From left to right, DreamBooth on Imagen, LoRA DreamBooth on Stable Diffusion, StyleDrop (round 1), StyleDrop (round 2, HF), StyleDrop (round 2, CLIP score feedback). A style reference image is shown in Fig. S8b.

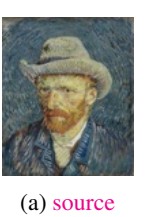

(a) source

(b) A baby penguin, animals

(c) A moose, animals

(d) A towel, artifacts

(e) An espresso machine, artifacts

(f) An avocado, produce and plants

(g) A crown, artifacts

Figure S11: Results comparison without cherry-picking. From left to right, DreamBooth on Imagen, LoRA DreamBooth on Stable Diffusion, StyleDrop (round 1), StyleDrop (round 2, HF), StyleDrop (round 2, CLIP score feedback). A style reference image is shown in Fig. S8c.

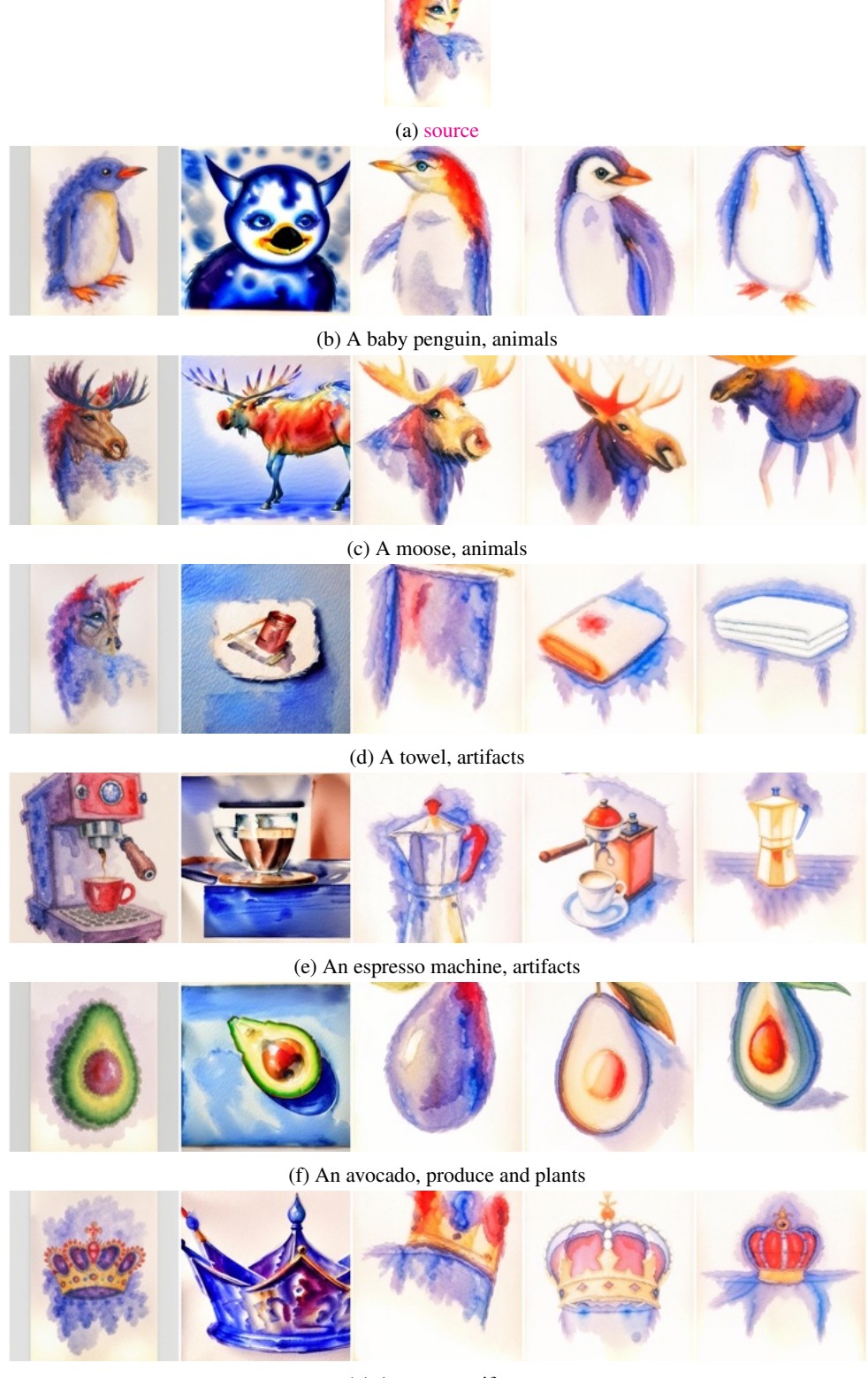

(a) source

(b) A baby penguin, animals

(c) A moose, animals

(d) A towel, artifacts

(e) An espresso machine, artifacts

(f) An avocado, produce and plants

(g) A crown, artifacts

Figure S12: Results comparison without cherry-picking. From left to right, DreamBooth on Imagen, LoRA DreamBooth on Stable Diffusion, StyleDrop (round 1), StyleDrop (round 2, HF), StyleDrop (round 2, CLIP score feedback). A style reference image is shown in Fig. S8d.

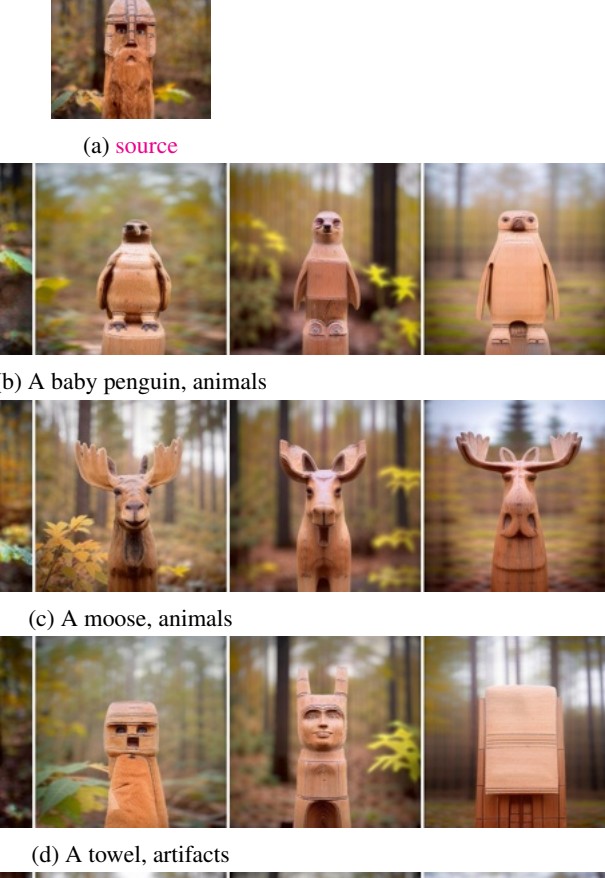

(a) source

(b) A baby penguin, animals

(c) A moose, animals

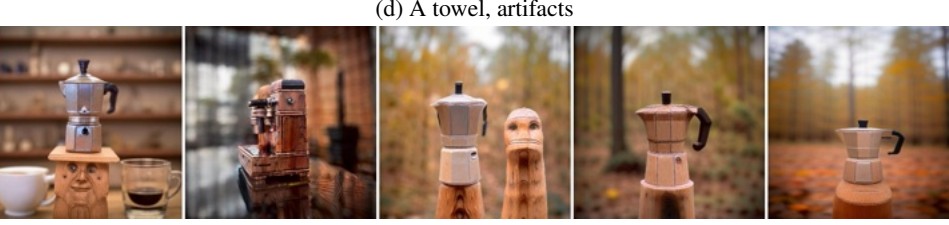

(d) A towel, artifacts

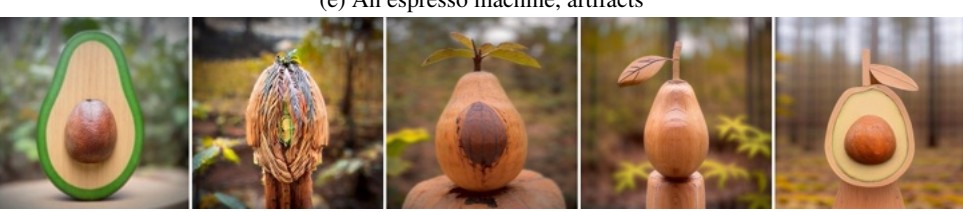

(e) An espresso machine, artifacts

(f) An avocado, produce and plants

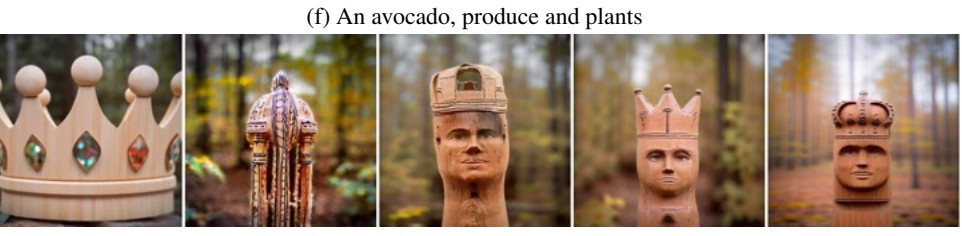

(g) A crown, artifacts

Figure S13: Results comparison without cherry-picking. From left to right, DreamBooth on Imagen, LoRA DreamBooth on Stable Diffusion, StyleDrop (round 1), StyleDrop (round 2, HF), StyleDrop (round 2, CLIP score feedback). A style reference image is shown in Fig. S8e.

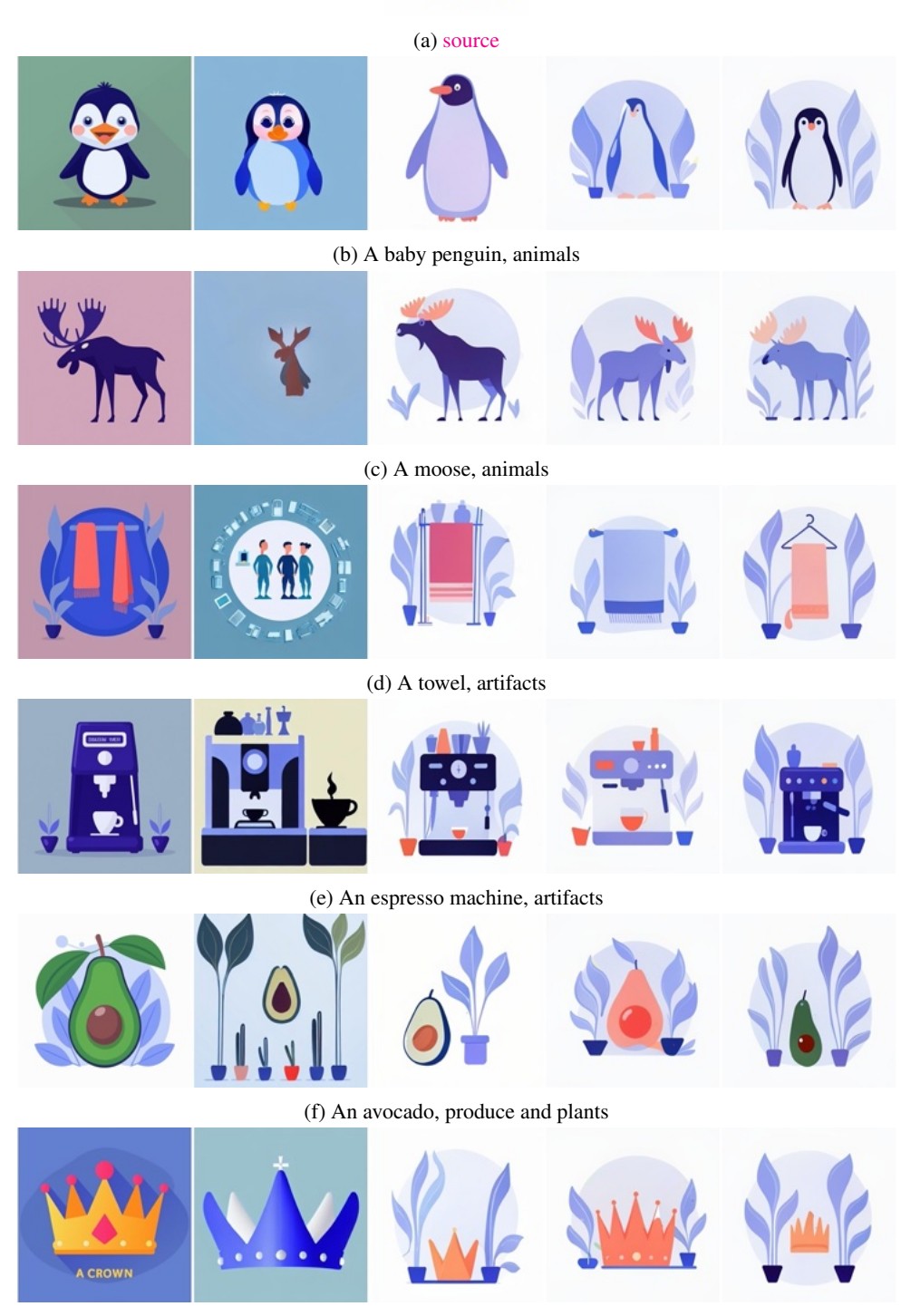

(a) source

(b) A baby penguin, animals

(c) A moose, animals

(d) A towel, artifacts

(e) An espresso machine, artifacts

(f) An avocado, produce and plants

(g) A crown, artifacts

Figure S14: Results comparison without cherry-picking. From left to right, DreamBooth on Imagen, LoRA DreamBooth on Stable Diffusion, StyleDrop (round 1), StyleDrop (round 2, HF), StyleDrop (round 2, CLIP score feedback). A style reference image is shown in Fig. S8f.

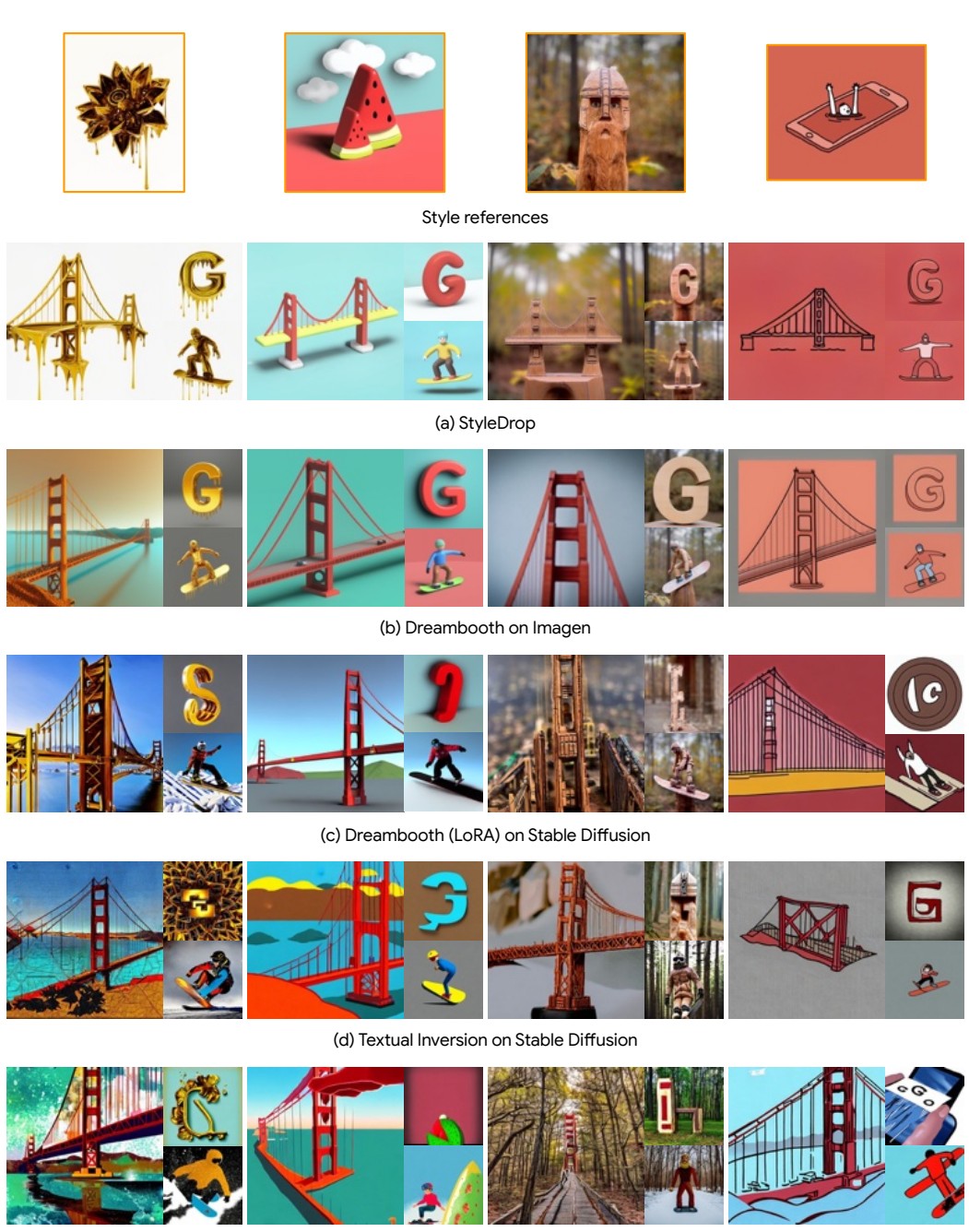

Figure S15: Style tuning comparison to baseline methods, including (b) DreamBooth on Imagen, (b) DreamBooth (LoRA) on Stable Diffusion, (c) Textual Inversion on Stable Diffusion, and (e) Hard Prompt Made Easy (PEZ) in Stable Diffusion.

