# OpenReview forum: "StyleDrop: Text-to-Image Synthesis of Any Style"
_NeurIPS.cc/2023/Conference — NeurIPS 2023 poster_

### Official Review · Reviewer_vFSY · 2023-06-28

**Soundness:** 3 good
**Presentation:** 3 good
**Contribution:** 3 good
**Rating:** 6
**Confidence:** 4

**Summary:**

This paper proposes a new method that conducts the image synthesis following a new style using a text-to-image model. The method is called StyleDrop, which works by efficiently learning a new style by fine-tuning few parameters, and improving the quality via iterative training with either human or automated feedback. StyleDrop is able to give impressive results even when the users supplies only a single image. Experiments show the effectiveness of the proposed method.

**Strengths:**

The experimental results, especially the user study results, demonstrate the effects of the proposed approach. The visual effects are good.

**Weaknesses:**

1.	This method needs iterative training with different feedbacks. Thus, I wonder whether the proposed strategy will sacrifice the finetuning efficiency even the parameter number to finetune is small. I want to see such comparisons.

2.	The strategy of adding trainable parameters in Fig.2 is similar to the ControlNet [A]. This method should be led into the comparison.

[A] Adding Conditional Control to Text-to-Image Diffusion Models, arxiv 2023

3.	Different human feedback may lead to different results which may depends on the professionalism of the users. I wonder the robustness of the proposed method towards different persons. Authors could provide the results with iterative feedback with different persons.


**Questions:**

Please see the weakness section and answer these questions.

**Limitations:**

This paper could provide the interactive experimental results with different persons and reference images.

---

> ### Author Rebuttal · Authors · 2023-08-09
>
> * Fine-tuning efficiency
>   * Training an adapter takes only 3 mins on TPU v3 with the batch size of 8 for 1000 steps for base transformer and 17 mins for super-res transformer for both rounds. Note that the super-res transformer does not need to be trained at every iterative training as it mostly serves for the super-resolution and the generation quality (e.g., text adherence) is strongly dependent on the base model. While the amount of time required for training increases, the requirement for compute or storage for saving model parameters remains efficient.
>
> * Comparison to [ControlNet](https://arxiv.org/abs/2302.05543)
>   * We agree that StyleDrop and ControlNet share similarity in that both are designed to learn additional model parameters. On the other hand, this idea is much more general than what is used in ControlNet, which may go back to the adapter tuning of language models ([Houlsby et al., 2019](https://arxiv.org/pdf/1902.00751.pdf)). Besides, ControlNet is trained on at least tens of thousands of images to work well (see Figure 22 of [[A](https://arxiv.org/abs/2302.05543)]) whereas StyleDrop works by learning from a single or a handful of style reference images. ControlNet has approximately 300M learnable parameters, while the number of adapter parameters is approximately 10M.
>
> * Impact from different human feedback
>   * We conduct experiments on 6 styles with 4 other human feedback. We observe some variance across different users’ feedback, which is expected. Such sensitivity to human feedback is important when the user's intention is expected to be reflected in generation (Section 4.4.4). Other than User 3, we observe an improved style score compared to using a CLIP feedback (0.673 Style score).
>
>   |             | Paper | User 1 | User 2 | User 3 | User 4 |
>   |-------------|-------|--------|--------|--------|--------|
>   | Style score ($\uparrow$) | 0.694 | 0.683  | 0.698  | 0.667  | 0.691  |
>   | Text score ($\uparrow$) | 0.322 | 0.328  | 0.326  | 0.322  | 0.327  |

---

> > ### Comment · Reviewer_vFSY · 2023-08-19
> >
> > Thanks for the rebuttal from the authors. I will keep my positive score.

---

### Official Review · Reviewer_oHZZ · 2023-07-01

**Soundness:** 4 excellent
**Presentation:** 4 excellent
**Contribution:** 3 good
**Rating:** 7
**Confidence:** 3

**Summary:**

The paper introduces a method to fine-tune a pre-trained large text-to-image model using a few images that provide an artistic style, so that the fine-tuned model can generate images that follow the provided artistic style. The proposed method is a combination of Muse [1], a transformer-based text-to-image generation model, and adapter tuning [2], a method to fine-tune a large text-conditioned transformer efficiently. The paper additionally proposes iterative training, i.e., using CLIP or human feedback to select a few good generated images and use them to train the proposed method for a second round or more rounds, to prevent leaked content from the style reference images. The method shows improved generation results over prior works such as DreamBooth [3]. The method can also be combined with DreamBooth [3] to generate images of a user-specified content with a user-specified style, while both the content and the style are provided as images.

[1] Muse: Text-to-image generation via masked generative transformers. arXiv preprint, 2023.
[2] Parameter-efficient transfer learning for nlp. International Conference on Machine Learning, 2019.
[3] DreamBooth: Fine tuning text-to-image diffusion models for subject-driven generation. CVPR, 2023.

**Strengths:**

- The paper is well-written. The preliminary works are introduced in the paper so I can understand the technical details even though I am not familiar with this field. The figure illustrations are also very helpful for me to quickly grasp the ideas behind this work.

- The qualitative results are astonishing.

- The comparison experiments and ablation studies are extensive with detailed analysis.

- The proposed method could potentially enable many applications, especially when coupled with DreamBooth to provide extra user control.

**Weaknesses:**

I do not find any significant weaknesses. Minor weaknesses are described below.

- I am not familiar with this field, therefore it is hard for me to comment on the novelty of this method. To me, it looks like a system built by combining Muse [1] and adapter tuning [2], with an optional iterative training step. It shows great results, but according to Figure 4 (b), the results of prior work DreamBooth are not bad at all, so I guess the performance improvement is marginal?

- One possible weakness is the training time. I do not find any mention of training time in the paper, so I assume it is going to take a long time to train a model. Therefore, the iterative training steps that require human feedback is going to be very inconvenient. The proposed method is unlikely to support interactive artistic design in real applications.

- I think the quantitative results may not be very informative. According to Table 2, the "Round 1" model without iterative training consistently outperforms other methods in terms of visual style alignment. The authors explained the possible reasons, but still, this can only mean one of two things: either the iterative training is unnecessary and the "Round 1" model without iterative training outperforms the full model, or the quantitative results are not informative and it is unclear whether the proposed method performs better than prior works such as DreamBooth.

**Questions:**

1. What is the training time for each round?

2. Any comments on the uninformative quantitative results?

**Limitations:**

Limitations and societal impact are discussed in the paper. They look fair.

---

> ### Author Rebuttal · Authors · 2023-08-09
>
> * DreamBooth is not bad at all, so I guess the performance improvement is marginal?
>   * We observe significant style score improvement with StyleDrop over DreamBooth in Tab 2 (0.644 → 0.705). Moreover, the user preference study clearly shows that the StyleDrop (86%) generates much more style consistent images than DreamBooth (9.7%). We believe that the performance gap is significant overall.
>
> * Training time
>   * Training an adapter takes only 3 mins on TPU v3 with the batch size of 8 for 1000 steps for base transformer and 17 mins for super-res transformer for both rounds. Note that the super-res transformer does not need to be trained at every iterative training as it mostly serves for the super-resolution and the generation quality (e.g., text adherence) is strongly dependent on the base model.
>   * Nonetheless, the proposed method, in its current form, wouldn’t support “real-time” interactive applications. Given recent progress on fast DreamBooth (e.g., [HyperDreamBooth](https://hyperdreambooth.github.io/), [taming encoder](https://arxiv.org/abs/2304.02642)), we believe that the StyleDrop can be further enhanced to support real-time interactive applications.
>
> * Quantitative metrics
>   * We agree that the Style and Text scores based on CLIP are proxy at best. Especially, they do not serve their purpose in isolation and should be understood in combination. For example, Style score can be “arbitrarily” close to 1.0 if the model is overfitted to generate a style reference image regardless of the text prompt (i.e., model collapse). On the other hand, such degenerate cases can be captured if we look at the text alignment score, as the text alignment score would be very low if the model collapses.
>   * As mentioned in line 139 -- 144, iterative training is to resolve the overfitting to a content and improve the text adherence, not necessarily to further improve the style learning capability. Table 2 clearly shows such a trade-off.

---

> > ### Comment · Reviewer_oHZZ · 2023-08-12
> > **Response to rebuttal**
> >
> > Thanks the authors for providing rebuttal.
> >
> > The rebuttal does not resolve my concerns on quantitative metrics.
> >
> > The authors claim in the rebuttal that "We observe significant style score improvement with StyleDrop over DreamBooth in Tab 2 (0.644->0.705)", but the text score drops from 0.335->0.313, right? As said in the rebuttal, "they (style and text scores) do not serve their purpose in isolation and should be understood in combination."
> >
> > From Table 2, after IT, the text score increases (0.313->0.322) and the style score drops (0.705->0.694), so I think it is reasonable to assume the IT step sacrifices style score to increase text score. In that case, why DreamBooth (text 0.313->0.335, style 0.705->0.644) is considered to be underperforming?
> >
> > I still do not think the quantitative metrics are very informative in comparison experiments. I will keep my original rating.

---

### Official Review · Reviewer_xBBj · 2023-07-04

**Soundness:** 4 excellent
**Presentation:** 3 good
**Contribution:** 3 good
**Rating:** 6
**Confidence:** 4

**Summary:**

To synthesize arbitrary image styles, this paper proposed a StyleDrop method that enables an image synthesis with the style of a given image, using text-to-image models, e.g., Muse and Imagen. StyleDrop consists of three crucial components: 1) a transformer-based text-to-image generation model using Muse, 2) an adapter tuning depended on [32]; and 3) an iterative training with CLIP and Human feedbacks. Extensive results demonstrate the effectiveness of the proposed StyleDrop method, and show that StyleDrop on Muse is better than the other related methods, including DreamBooth and textual inversion on Imagen or Stable Diffusion.

**Strengths:**

1.  The main strength is that the extensive visual results are impressive and show the nuances and details of a user-provided style.

2. This paper proposed an incremental method to synthesize arbitrary image styles as it uses CLIP and human feedbacks to train a new adapter in the Muse model for the stylized text-to-image synthesis.

3. This paper is well-written and is easy to follow.

**Weaknesses:**

1. Since the proposed StyleDrop model is based on Muse, the reviewer thinks the Muse should be as a baseline. However, their visual comparison is missing expect the CLIP scores in Table 2. This is unnormal. Please show some visual results.

2. Why does this paper construct the form of Eq. (4)? How about the following forms?

  1) l_k=\hat{G}(v_k,T(t),theta)+lambda_A(\hat{G}(v_k,T(t),theta)- G(v_k,T(n)))
  2) l_k=\hat{G}(v_k,T(t),theta)+lambda_A(\hat{G}(v_k,T(t),theta)-0.5*(G(v_k,T(t))+G(v_k,T(n))))

It sees that there is only one hyper-parameter in the above two forms.

3. Overall, the framework is unclear although Fig. 2 shows the improved architecture. As the StyleDrop is an improved Muse model, please draw the overall pipeline so that readers and reviewers can better understand this work and the difference between StyleDrop and Muse.


**Questions:**

Please see the weaknesses.

**Limitations:**

Yes

---

> ### Author Rebuttal · Authors · 2023-08-09
>
> * Visual comparison to Muse baseline
>   * Please find a visual comparison to the Muse baseline in Figure R1 of the attached rebuttal PDF in the global response. Generated images by Muse and StyleDrop share the same text prompt (e.g., "A baby penguin in watercolor painting style"), but we see that their styles are very different. StyleDrop generates a style consistent with the style in the reference image, but Muse baseline does not. This is in line with Style scores reported in Tab 2. (Muse baseline: 0.556, StyleDrop (HF): 0.694).
>
> * Classifier-free guidance
>   * As mentioned in line 123 -- 125, the design of our logit sampler is to provide flexibility in controlling the adaptation (i.e., style) and text. We ablate $\lambda_{A}$ and $\lambda_{B}$ in Section B.3.1 and Figure S4 in the supplementary material to show how these two values interact at generation.
>   * We note that the CFG formulations suggested by the reviewer are special cases of our formulation, when 1. $\lambda_{A} = \lambda_{B}$ and 2. $0.5 * \lambda_{A} = \lambda_{B}$. As such, we conducted experiments with the reviewer's suggested parameters of 1. $\lambda_{A}=5.0$ and 2. $\lambda_{A}=10.0$, while fixing $\lambda_{B}=5.0$. Note that $\lambda_{A}=2.0$ was used in our experiments. We report the style and text scores based on CLIP in the table below. We observe a clear style-text score trade-off as we change $\lambda_{A}$.
>
>   |             | $\lambda_{A}=2.0$ | $\lambda_{A}=5.0$ |  $\lambda_{A}=10.0$    |
>   |-------------|-------------------|--------------------|-------|
>   | Style score ($\uparrow$) | 0.705             | 0.709              | 0.716 |
>   | Text score ($\uparrow$) | 0.313             | 0.308              | 0.303 |
>
> * StyleDrop framework
>   * We've included the overall framework of StyleDrop and how it fits within the Muse framework in the rebuttal PDF (Figure R3). Notably, the Muse is composed of 1) image tokenizer and detokenizer that converts between pixel image and a discrete token sequence, 2) T5 text embedding, 3) and transformer layers that models the distribution of a discrete token sequence conditioned on the text prompt. Our adapter tuning framework tackles the fine-tuning of transformer layers, and Figure 2 describes how to add an adapter to each transformer layer.

---

> > ### Comment · Reviewer_xBBj · 2023-08-21
> >
> > Thank you for the authors' rebuttal. I will maintain my positive score.

---

### Official Review · Reviewer_bhPK · 2023-07-05

**Soundness:** 3 good
**Presentation:** 3 good
**Contribution:** 3 good
**Rating:** 6
**Confidence:** 4

**Summary:**

This paper works on personalized text-to-image synthesis. Given reference style images (even a single one), the model can adapt the style to the synthesized images. There are several key components in the papers including: using a transformer-based t2i model rather than a diffusion-based one; adding an adaptor into the model for training; incorporating human-in-the-loop feedback for selecting useful training data during iterative learning. Extensive details with experiments have been included to validate the effectiveness of the proposed method.

**Strengths:**

+ The problem of adapting a t2i model based on a single style image is interesting and worth studying. It should also be valuable to this venue.
+ The proposed method is well-motivated, and the components seem effective based on the experiments.
+ The extensive supplementary and limitation discussion is appreciated, especially for the interface for collecting human feedback.
+ The paper generally writes well and easy-to-follow.
+ The results look great.

**Weaknesses:**

Overall, I am kind of on the supportive side but with several questions/concerns.

+ Reproducibility: It would be great if more context on reproducibility is provided. Will the code be released? If the code cannot be released in a complete way, how the reproducibility be achieved?
+ Generability: The paper argues that using MUSE - a transformer-based method is more reasonable, but the proposed components such as the adaptor and human feedback can be used in other models such as diffusion. What will the performance be if all these proposed components are used on the other frameworks?
+  How large effect the negative prompt will be, and it would be great if there are some specific examples or statistics that can be shown.
+ It's still not clear to me why the model only captures the style rather than the content. is this influenced by the human-selected images? Can the method be directly adapted to learning concept/content just like dream-booth? If not, may I know what factors let the model fail to capture the content/semantic info of the references?
+ In L120, it is saying "we use different text prompts for each input image to better disentangle ...". I am not sure whether I understand this sentence or not. Besides, is there any ablation study related to this claim?
+ In L161, how many images are used for annotation? When compare with the baseline, especially with single reference images, are the human feedback also incorporated? if that's the case, the experiment may be unfair, as though those images are generated by the model itself, incorporating human feedback should also be considered as labeling/more reference data.
+ How robustness of the model is unclear, how many annotated images are necessary to maintain the high-quality. Will the number vary a lot for different cases? Besides, the metrics are only computed on a few of styles which may not be enough or conclusive. More discussions should be added here to evaluate the paper fairly rather than simply saying they are the future works.

**Questions:**

Please see the comments above.

**Limitations:**

The limitation has been studied but is not that convincing. It would be great to show some specific failure cases in the paper.

---

> ### Author Rebuttal · Authors · 2023-08-09
>
> * Reproducibility
>   * We'll provide more complete pseudo-code on the training and inference of the StyleDrop, in addition to what we've already provided in the paper (e.g., dataset, hyperparameters, pseudo-code on generating adapters). In addition, we'll look into options to release the StyleDrop implementation based on open-source Muse models (e.g., [open-muse](https://github.com/huggingface/open-muse), [muse-pytorch](https://github.com/baaivision/MUSE-Pytorch)).
>
> * Generability
>   * Some components of StyleDrop (e.g., adapter tuning, iterative training) are model agnostic and can be integrated into diffusion model fine-tuning. We've provided results for LoRA on Stable Diffusion, which is also a powerful parameter-efficient fine-tuning method, similar to the adapter tuning. For iterative training, we point out our ablation experiments in Section 4.4.1 to the reviewer. See Figure 4(d), where we fine-tune an Imagen on 10 generated images of StyleDrop (round 1) on Muse using a descriptive style descriptor. While we see an improvement, we still find that the diffusion-based models fall short of Muse in learning style when following fine-tuning recipes of existing works (e.g., DreamBooth, LoRA Stable Diffusion). Given a better performance with StyleDrop (round 1) on Muse over DreamBooth on Imagen, this result would serve as a performance upper bound of iterative training of diffusion models.
>
> * Effect of negative prompt
>   * We use the negative prompt of "Shutterstock, watermark", hoping to remove watermarks in generation, in place of unconditional distribution, following [Muse](https://arxiv.org/pdf/2301.00704.pdf) (see Section 2.7).
>
> * Model captures style rather than content
>   * Content and style disentanglement is done by constructing a text prompt that is composed of content and style descriptors, as described in Section 3.2.1. By replacing a content descriptor at generation, we can generate an image of a different content but with a desired style.
>   * However, this does not mean that the model is failing at capturing the content -- indeed, it also captures the content. As evidence, one common failure case of StyleDrop (round 1) model is a "content leakage" (e.g., Figure 3, same house appears at generation). In addition, as shown in Section 4.3, the same adapter tuning method can be used for subject tuning (i.e., DreamBooth) on Muse, leading to a generation of "my subject in my style". In this case, the content descriptor (e.g., a teapot, a vase) should remain the same between training and generation.
>
> * Different text prompts for each input image
>   * Note that we construct text prompts as a composite of content and style descriptors (e.g., Table 1, Table S4). Unless different training images contain the same content and style, text prompts of different input images would have different text prompts. This is different from the DreamBooth where there is a unique subject and text prompts can be shared across different input images. On the other hand, this is a natural design choice for StyleDrop with iterative training, where we train a model with multiple generated images using different content descriptors. When learning from a single image, they do not differ.
>
> * How many images are used for annotation?
>   * Note that the method is fine-tuned on a single style reference image for "round 1" and multiple generated images for "round 2". We refer Tab. 2 to the user, where different fine-tuning methods using a single (round 1) style reference images are **fairly** compared. For example, “DB on Imagen” and “StyleDrop on Muse (Round 1)” are both fine-tuned on a single image, so the comparison should be fair. Figure S8 – S13 has a visual comparison of these methods.
>
> * How robust?
>   * As mentioned, the model can be learned from a single image quite reliably (e.g., all 24 StyleDrop models shown in the paper are learned from a single style reference image). We shared generation results on all 24 models **without cherry-picking** (e.g., all images are sampled using the same random seed) in our [anonymized project page](https://styledrop.github.io/anon), which was submitted as part of a supplementary material.
>
> * Limitation / failure cases
>   * We visualize generated images **without cherry-picking** (i.e., images are generated with the same random seed) in Figure S8 -- S13 as well as our [anonymized project page](https://styledrop.github.io/anon), submitted as part of a supplementary material. We also discuss challenging cases in the rebuttal PDF, Figure R2, where content - style disentanglement gets more difficult when the content of the style reference image cannot be clearly described. In such cases (Figure R2 in rebuttal PDF), we find that the method still struggles to disentangle style and content even after a round of human feedback and requires multiple rounds of iterative training.

---

### Official Review · Reviewer_9biG · 2023-07-27

**Soundness:** 2 fair
**Presentation:** 3 good
**Contribution:** 2 fair
**Rating:** 4
**Confidence:** 4

**Summary:**

This paper proposes StyleDrop to perform text-to-image generation following the style of a reference image. StyleDrop finetunes Muse [1] via parameter-efficient fine-tuning (PEFT) using the style reference images and their corresponding text prompts. To better finetune Muse, this paper proposed (1) a templated approach to construct the text prompts for style reference images, (2) iterative training with feedback (model-based or human-based). This paper also tried to combine DreamBooth and StyleDrop, yeilding personalization of both style and content.

Ref:
[1] Huiwen Chang et al. Muse: Text-to-image generation via masked generative transformers.

**Strengths:**

(1) StyleDrop is able to produce high-quality and visually pleasing synthesis results, even when the style images and the content text prompt are highly unrelated.

(2) This paper presents a well-established system which leverages both a large model (Muse) and human feedback to handle this task.

(3) The overall presentation is satisfying, and the paper is easy to read.

**Weaknesses:**

(1) There is no significant technical contribution in this paper. Section 3.2.1, 3.3, and 3.4 are more like engineering tricks to me. The task of style tuning text-to-image models is also not new, given the existence of DreamBooth, Custom Diffusion, etc.

(2) In the data-collection section (Line 185-188), what is the criterion of selecting styles? The authors did not make it clear. A skeptical view is that the authors could select the styles that favor their method.

(3) Some important hyper-parameter and ablation studies are missing. For example,

(a) how the number of style reference images will affect the model performance? (I assume that StyleDrop is not overfitting to just one style.)

(b) When comparing StyleDrop on MUSE to DreamBooth on Imagen/ Stable Diffusion, how much performance gain is from MUSE? Can DreamBooth be applied in Muse? It seems unfair to compare StyleDrop on Muse vs DreamBooth on Imagen/Stable Diffusion.

**Questions:**

(1) How many style reference images are needed to finetune Muse? In Line 127, the number of $N$ seems not clear. Is $N$ the number of images in Table S1? How the number $N$ will affect the model performance seems not discussed.

(2) When a new style reference image is added, how to finetune StyleDrop to fit it? Using only this style image or all style images so far?

**Limitations:**

The authors did not show any failure cases of StyleDrop. In Line 321-327, they mentioned the visual styles are more diverse than what is possible to explore in the paper. It is necessary to conduct some experiments to show what StyleDrop is capable of and not capable of. I don’t think StyleDrop can always generate good results for any given style references.

---

> ### Author Rebuttal · Authors · 2023-08-09
>
> * Technical contribution
>   * We note that the most previous works (e.g., [DreamBooth](https://dreambooth.github.io/), [Custom Diffusion](https://github.com/adobe-research/custom-diffusion)) on t2i adaptation are based on diffusion models, and no framework has been proposed for t2i generative vision transformers, such as [Muse](https://muse-model.github.io/). To our knowledge, we are the **first** to formulate a few-shot fine-tuning framework for t2i generative vision transformer and demonstrate impressive generation results with style-tuning.
>   * While style-tuning has been studied in previous works, we pushed the envelope significantly by learning from a **single** style reference image. This is made possible by two innovative and non-trivial designs that enhance disentanglement of content and style in the style reference image and resolve the limitation of one-shot fine-tuning of t2i models, with the carefully designed text prompt (Section 3.2.1) and iterative training with human feedback (Section 3.3). Furthermore, we propose to sample from two distributions (Section 3.4) to allow the generation of “my subject in my style” via subject- and style-tuned adapters.
>
> * Data collection
>   * We selected 24 styles from 6 style categories -- watercolor painting, oil painting, line drawing / flat illustration, lighting condition, 3d rendering, sculpture. To our knowledge, this is the largest number of styles considered in a single paper in this domain. For example, [Textual Inversion](https://arxiv.org/pdf/2208.01618.pdf) and [Custom Diffusion](https://arxiv.org/pdf/2212.04488.pdf) have shown on 2 styles each. We did conduct experiment with more style reference images internally and the model did work well on most cases, but didn’t include in the paper as 1) we think 24 styles are enough and 2) due to lack of space and time. We’ll include more styles as needed in the supplementary material of the final version.
>   * Our selection of images covers diverse visual styles, and yet each style reference image is chosen to be unique so that the effectiveness of style learning can be easily verified. We tried to avoid well known artworks as much as possible (except for those of Van Gogh) as they are likely known by t2i models already and as such they are not good examples to verify the fine-tuning and adaptation capability. In addition, we select images with a proper license.
>
> * Limitation / failure cases
>   * We visualize generated images **without cherry-picking** (i.e., images are generated with the same random seed) in Figure S8 -- S13 as well as our [anonymized project page](https://styledrop.github.io/anon), submitted as part of a supplementary material. We also discuss challenging cases in the rebuttal PDF, Figure R2, where content - style disentanglement gets more difficult when the content of the style reference image cannot be clearly described. In such cases (Figure R2 in rebuttal PDF), we find that the method still struggles to disentangle style and content even after a round of human feedback and requires multiple rounds of iterative training.
>
> * Number of style reference images? When a new style reference image is added?
>   * We clarify that the StyleDrop learns **a separate model** for each style, from a small amount (e.g., 1) of images consistent in that style. In other words, the StyleDrop (Round 1) model is trained per style from a single (N=1) style reference image. For iterative training, we use N=10 generated images per style.
>
> * Comparison of StyleDrop on Muse (ref: Table 2) vs DreamBooth on Muse, both without iterative training:
>     |             | StyleDrop on Muse | DreamBooth on Muse |
>     |-------------|-------------------|--------------------|
>     | Style score ($\uparrow$) | 0.705             | 0.654              |
>     | Text score ($\uparrow$) | 0.313             | 0.308              |

---

> > ### Comment · Reviewer_9biG · 2023-08-14
> > **Response to rebuttal**
> >
> > Thank the authors for the response.
> >
> > I am especially interested in the Comparison of StyleDrop on Muse vs DreamBooth on Muse. Could you show us more qualitative results (visual comparisons) in addition to the numbers you reported? For Text score (0.313	vs 0.308), the advantage of StyleDrop is very marginal. I need visual comparisions to fully unsterstand the advantages of StyleDrop.
> >
> > I think it is okay to send an anonymous link to the AC and then AC passes it to me.

---

> > > ### Author Response · Authors · 2023-08-14
> > > **Happy to share!**
> > >
> > > Thanks for your response. Happy to share more visual results upon AC's approval.
> > >
> > > In the meantime, while text adherence is an important metric, we emphasize that the advantage of StyleDrop is at the improved style consistency, which is clearly shown by the Style score (StyleDrop: 0.705 vs DreamBooth: 0.654), without losing text adherence. Again, we are more than happy to share visual examples if AC allows, but the Style score itself should be able to tell the performance difference already.

---

### Author Rebuttal · Authors · 2023-08-09

We thank reviewers for reviewing our submission. We are encouraged with positive feedbacks on the **quality of image generation results** (9biG, bhPK, xBBj, oHZZ, vFSY), **extensive experiments** (bhPK, xBBj, oHZZ, vFSY), **well-motivated method** (bhPK, xBBj), and **well-written manuscript and presentation** (9biG, bhPK, xBBj, oHZZ).

In the rebuttal PDF, we included 3 figures:

* Figure R1: Visual comparison between Muse baseline and StyleDrop (reviewer xBBj).
* Figure R2: Limitation of StyleDrop when style reference image cannot be easily described as content + style descriptors (reviewer 9biG, bhPK).
* Figure R3: StyleDrop framework. Adapter (A) is included to each layer of base (and super-res) transformer of Muse (reviewer  xBBj).

In addition to the rebuttal PDF, we encourage reviewers to take a look at our [anonymized project page](https://styledrop.github.io/anon), submitted as part of a supplementary material in the initial submission. We shared generation results on all 24 models **without cherry-picking** (e.g., all images are sampled using the same random seed). Also, Figure S8 – S13 in the supplementary material of the initial submission include a visual comparison between DreamBooth on diffusion models and StyleDrop on Muse, of which images from StyleDrop on Muse are shown **without cherry-picking** (e.g., all images are sampled using the same random seed). As such, we see some failure cases (e.g., Figure S8 (d) third column shows a towel without changing the style), but then are fixed with iterative training with human or CLIP feedback.

We respond to each reviewer below to address concerns. Please take a look and let us know if further clarification / discussion is needed.

---

### Comment · Area_Chair_eY1x · 2023-08-15
**Please read the authors' responses and start discussing.**

Hi reviewers,

The authors have responded to your reviews. Please carefully read their responses, consider to what extent they address your concerns, and update your review accordingly. If you have follow-up questions to anything the authors have written in their responses, I encourage you to reply to them and continue the discussion.

Your AC

---

### Decision · Program_Chairs · 2023-09-21

**Decision:**

Accept (poster)

**Comment:**

Most reviewers (4 out of 5) recommend accepting this paper (1 “Accept”, 3 “Weak Accept” and 1 “Borderline reject”). All reviewers mentioned that the results of this paper are impressive. The AC agrees with the reviewers that the paper is interesting and deserves to be published in NeurIPS 2023.